# How hard do avalanche practitioners tap during snow stability tests?

Håvard B. Toft[1,2], Samuel V. Verplanck[3] and Markus Landrø[1,2]

[1] The Norwegian Water Resources and Energy Directorate, Oslo, Norway
[2] Center for Avalanche Research and Education, UiT The Arctic University of Norway, Tromsø, Norway
5  [3] Department of Mechanical Engineering, Montana State University, Bozeman, MT, USA

*Correspondence to*: Håvard B. Toft, Norwegian Water Resources and Energy Directorate, Oslo, Norway; tel: +47 454 82 195; email: htla@nve.no

**Abstract.** This study examines the impact force applied from hand taps during Extended Column Tests (ECT), a common method of assessing snow stability. The hand-tap loading method has inconsistencies across the United States, Canadian, 10  Swiss, and Norwegian written standards, as well as inherent subjectivity. We developed a device, the "tap-o-meter", to measure the force-time curves during these taps and collected data from 286 practitioners, including avalanche forecasters and mountain guides in Scandinavia, Central Europe, and North America. The mean, median, and inner quartile peak forces are distinctly different for each loading step (wrist, elbow, and shoulder), and the peak force approximately doubles from one loading step to the next. However, there is considerable overlap across the range of measurements and examples of participants with higher 15  force wrist taps than other participants' shoulder taps. This overlap challenges the reliability and reproducibility of ECT results, potentially leading to dangerous interpretations in avalanche decision-making, forecasting and risk assessments. Our results provide an answer to the question of "How hard *do* avalanche practitioners tap?" but not necessarily "How hard *should* avalanche practitioners tap?". These data and insights are intended to facilitate discussion among the tests' creators, the scientific community, and the practitioner community to update thresholds, guidelines, and test interpretation.

20  ## 1 Introduction

Snowpack instability describes the propensity for a slope to avalanche (Reuter & Schweizer, 2018). Failure initiation and crack propagation are key components of the avalanche release process (Reuter & Schweizer, 2018). Stability tests[1] help gather crucial information on weak layer identification, failure initiation, and crack propagation. Determining snowpack stability is a core concept in avalanche forecasting and backcountry decision-making, yet it is a challenging measure to quantify. In 25  backcountry travel, the decision process ultimately ends with a go or no-go decision based on an assessment of avalanche likelihood, avalanche size, and potential consequences. Snowpack stability evaluation is essential in assessing avalanche likelihood in such a context. To aid this complex decision-making process, snow stability tests can support decision making in case of conditional stability (e.g., Birkeland et al, 2023). They provide a structured analytical approach, particularly valuable when direct signs of instability, like recent avalanches, shooting cracks, or whumpfs, are absent.

30

---

[1] In our paper, we will often use terms "snowpack stability" and "stability tests", rather than "snowpack instability" and "instability tests", due to their widespread usage in the avalanche practitioner community.

In contrast, in situations with poor snowpack stability, nature provides apparent signs such as recent avalanches, shooting cracks, and "whumpfs". These clear signs of instability are commonly referred to as Class I factors (instability factors) in a three-class division based on informational entropy (LaChapelle, 1980; D. McClung & Schaerer, 2006). The more stable the snowpack, the greater the load it can support before it fails. The instability can be less evident in these situations, and more indirect factors such as stability tests (class II) and meteorological factors (class III) should be evaluated. Hence, stability tests can be of great importance in avalanche forecasting and provide highly valuable information to the backcountry traveler.

One of the first documented field snow tests is the shovel shear test developed by Faarlund and Kellermann in 1974 (originally known as the Norwegermethode; Kellermann, 1990). Although the role of compressive stress in weak layer failure was in discussion at the time (Perla & LaChapelle, 1970), weak layer shear strength - measured with a shear frame – was a typical metric for slope stability, and the shovel shear test provided a convenient field method of obtaining similar information.

In the late 1980s, Föhn (1987) quantified the Rutschblock (RB) test into the seven known levels today. In the 1990s, the compression test (CT) became popular (Clarkson, 1993; Jamieson & Johnston, 1996). Both the CT and RB involve loading the snow surface, transmitting stress through the slab, and the possibility of weak layer failure. A distinction between these tests lies in their load application method: the CT utilizes hand-taps, while the RB test requires the load of a person on skis.

The propensity for an initiated crack to propagate became a popular concept as a collapse-based, crack-propagation model (Heierli et al., 2008) had conflicting results with a shear-based, crack-propagation model (D. McClung, 1979). In line with this discussion, the propagation saw test (PST) (Gauthier & Jamieson, 2008, 2006) and extended column test (ECT) (Simenhois & Birkeland, 2006) were developed as field tests to assess propagation propensity. The ECT is a frequently used test by avalanche practitioners and recreationists. The test has been validated in different geographies and avalanche climates such as continental and intercontinental climates of the United States (Birkeland & Simenhois, 2008; Hendrikx & Birkeland, 2008; Simenhois & Birkeland, 2009), the Swiss Alps (Techel et al., 2020; Winkler & Schweizer, 2009), the Spanish Pyrenees (Moner et al., 2008) and New Zealand (Hendrikx & Birkeland, 2008; Simenhois & Birkeland, 2006).

The four stability tests described above measure different types of information in the snowpack using different triggering mechanisms, set-ups, and dimensions. Relevant types of information are whether the test can (1) identify weak layers in combination with slabs, (2) measure failure initiation, and (3) measure crack propagation. We have summarized the properties of each test in Table 1 with inspiration from Birkeland et al. (2023).

**Table 1: Different types of information that can be extracted from the four different stability tests (modified from Schweizer and Jamieson, 2010; Birkeland et al. 2023).**

| Test | Identifying weak layer below slab | Measures failure initiation | Measures crack propagation | Triggering mechanism | Dimensions (width, upslope) |
|------|-----------------------------------|-----------------------------|----------------------------|----------------------|------------------------------|
| RB | Yes | Yes | Yes | Weight of a human | 2 m x 1.5 m |
| CT | Yes | Yes | Partly | Hand-tap | 30 cm x 30 cm |
| ECT | Yes | Yes | Yes | Hand-tap | 90 cm x 30 cm |
| PST | No | Partly | Yes | Cutting with saw | 30 cm x 100 cm[1] |

[1] or the weak layer depth, whatever is greater.

As is evident in Table 1, stability tests are meant to reflect the avalanche release process. To connect stability tests with slope-wide avalanche mechanics, a mathematical model of the stability test is needed. To date, most of this modeling has been done with the PST (Benedetti et al., 2019; McClung & Borstad, 2012; Van Herwijnen et al., 2016; Weißgraeber & Rosendahl, 2023). A key component of the CT and ECT is the hand-tap loading which creates a boundary condition for a mathematical model of the CT and ECT. Creating this model is out of our scope, however, characterizing the impact curves is an important step towards modeling the CT and ECT.

To conduct an ECT, the hand-tap loading method is implemented that was originally developed for the CT. There are subtle differences in the current guidelines for these hand-taps. The American Avalanche Association (2022) defines the most recent US standard as follows. This is similar to the Canadian standard (Canadian Avalanche Association, 2016), which has expanded the definition by including the text marked with *italics*.

1. "Tap 10 times with fingertips, moving hand from wrist."
2. "Tap 10 times with the fingertips or knuckles moving forearm from the elbow. *While moderate taps should be harder than easy taps, they should not be as hard as one can reasonably tap with the knuckles*".
3. "Hit the shovel blade moving the arm from the shoulder 10 times with open hand or fist. *If the moderate taps were too hard, the operator will often try to hit the shovel with even more force for the hard taps - and may hurt his or her hand*".

In other countries, the instructions vary as well. For example, in Switzerland the instructions are simply described using a single sentence: «*The blade of the avalanche shovel is placed on the block on one side and successively loaded with 10 hits each from the wrist (1-10), the elbow (11-20) and the shoulder (21-30).* » (Dürr and Darms, 2016). There are further discrepancies if we look at the Norwegian standard (Norwegian Water Resources and Energy Directorate, 2022).

"For every sequence of 10 taps, the load is increased as follows:

1. Let the hand fall with its own weight, lifted from the wrist.
2. Let the hand and forearm fall with their own weight, lifted from the elbow.
3. Let the entire arm fall with its own weight, using a fist, lifted from the shoulder."

If a failure in the snowpack is detected during any of the taps, the specific tap number along with the depth of the weak layer is recorded for further investigation. For example, if a failure propagates at the 21st tap at a depth of 40 cm, it would be noted as 'ECTP21@40cm. The interpretation of ECT test results remains an open discussion. Originally, a binary interpretation of test results was suggested, referred to as $ECT_{orig}$ in this paper. Specifically, if a fracture initiates but does not propagate (ECTN), then the test result is considered stable. In contrast, if a fracture propagates across the extended column (ECTP, or ECTPV if during isolation), then the test result is considered unstable. If no fracture is initiated within the 30 taps, the outcome is neither stable nor unstable, and should therefore be regarded as inconclusive.

Another classification was suggested by Winkler and Schweizer in 2009 ($ECT_{w09}$), using three classes divided by the number of taps needed to initiate a fracture with or without propagation:

- ECTP ≤21 – low stability
- ECTP >21 – intermediate stability
- ECTN or ECTX – high stability

Recent work by Techel et al. (2020) ($ECT_{t20}$) suggests using four classes and applying the established labels for snow stability: poor, fair and good (e.g. American Avalanche Association, 2022):

- ECTP ≤13 – poor
- ECTP >13 to ECTP ≤22 – poor to fair
- ECTP >22 or ECTN ≤10 – fair
- ECTN >10 or ECTX – good

The variability in tapping force has been a known limitation for the CT and ECT interpretation (American Avalanche Association, 2022; Schweizer & Jamieson, 2010; Techel et al., 2020). Birkeland and Johnson (1999) attempted to remedy this limitation by developing the stuffblock test. The test uses a nylon sack filled with ~4.5 kg (10 pounds) of snow which is dropped on a CT or ECT column with 10 cm increments until a failure initiation is reached.

There have been some previous studies to measure the applied force of hand tapping, as well as quantify the stress-state within the snow during these loads. Logan (2006) made measurements of hand taps during a conference to learn more about timing, impact force and technique, but the results were never published. Thumlert and Jamieson (2015) impacted the snow with both a drop hammer and hand taps and measured the resulting stress within the snow. Our study expands on the work of Sedon (2021) and Griesser et al. (2023). Each of these studies measured tap force by avalanche practitioners (n=69 and n=62, respectively) in an indoor setting. Furthermore, Griesser et al. (2023) performed stress measurements during CTs in the field

and investigated the effects of body characteristics such as weight and height. Their analyses consist of bivariate tests, i.e., testing if people who are heavier tap harder, and if people who are taller tap harder. A limitation of this approach is that, since height and weight are typically correlated, the tests do not reveal which of the two factors that are more important, or if height (weight) affect tap force at a given weight (height). Regarding sampling rate, a critical aspect of accurately measuring dynamic loads, Sedon (2021) does not specify theirs and Griesser et al. (2023) use a sampling rate of 100 Hz (one measurement every 10 ms).

The objective of our work is to develop an improved measurement device with adequate sampling rate that can accurately characterize the impact curves of hand-tap loading and investigate the interpersonal variability between participants from different geographical regions. We plan to use multivariate regression to investigate whether body characteristics, snow climate, and gender influence the impact force from hand taps. Furthermore, we intend to not only measure the peak force, but also the loading rate, a metric not included in the previous studies by Sedon (2021) and Griesser et al. (2023). It has been well-established that snow's response depends on the loading rate (Shapiro et al., 1997), a quantity shown to both influence stress wave transmission through snow slabs (Verplanck and Adams, 2024) as well as failure of weak layers such as depth hoar, facets, and surface hoar (Reiweger et al. 2015). Thus, peak force alone is not enough information to accurately understand and predict snow's response dynamic loads. Determining how snow responds to the applied force from a hand-tap is outside of our scope, however, a quantified understanding of how hard practitioners tap will aid in the process of updating standards for test execution and interpretation.

## 2 Methods

### 2.1 The device: "tap-o-meter"

To measure the force from hand taps, a device dubbed the "tap-o-meter" was made. A total of three devices were built to enable data collection in different parts of the world in a similar time frame (Fig. 1). Each "tap-o-meter" has the following components:

1. A shovel blade which acts as the loaded surface.
2. A load cell to transduce the tapping force into an electric signal.
3. Oscilloscope with a voltage amplifier to measure the signal.
4. 30 x 30 x 0.6 cm stainless steel base to provide a sturdy foundation.

### 2.1.1 Load cell

A single, cantilever-style load cell from Load Cell Central (GCB3-SS-M-50KG) was used to measure the tapping force. The recommended capacity of the load cell is 490 N, with an ultimate overload rating of 1,470 N. The full-scale output (FSO) of the load cell is 2 mV/V and refers to the maximum output signal that the load cell can produce for its rated capacity.

### 2.1.2 Oscilloscope and voltage amplifier

An oscilloscope (Digilent Analog Discovery II) was used to measure the impact force. The oscilloscope provides a 5-volt input to the load cell, which yields a maximum output signal of 10 mV with the FSO from the load cell. The minimum change in voltage that can be measured by the oscilloscope is 0.2 mV. To increase the measurement resolution, a linear voltage amplifier was added between the load cell and the oscilloscope. The amplifier was custom built using an AD8429 amplifier from Analog Devices. The amplification, or gain (G), is controlled by an external two-pin resistor ($R_{ext}$), using the following equation:

$$G = 1 + \frac{6000\ ohm}{R_{ext}},\qquad\qquad(1)$$

In our study, we used a 30-ohm resistor, resulting in a 201x amplification of the output signal from the load cell. Using this setup, the oscilloscope is theoretically able to measure 10,050 steps between 0-490 Newtons, or 30,150 loading steps between 0-1,470 N. The device was calibrated statically by using a set of known weights ranging from ~50 to 300 N (Appendix-1), resulting in a linear regression with $R^2 = 0.999998$.

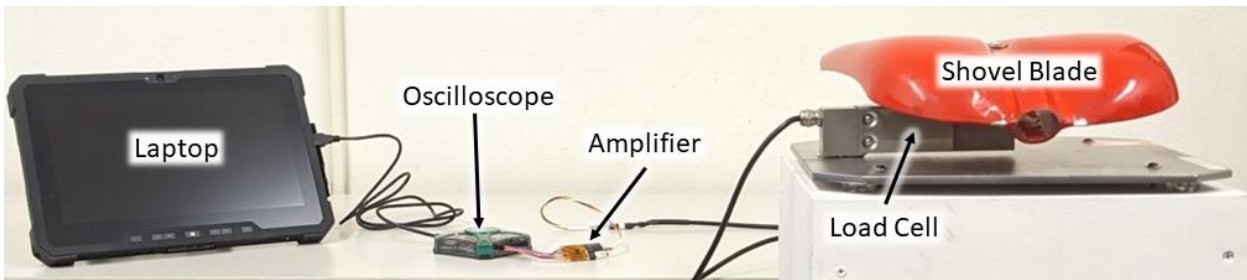

**Figure 1: The "tap-o-meter" consists of a metal base with the load cell and shovel blade attached above. The load cell is connected to the oscilloscope through the custom-built 201x amplifier.**

To determine an appropriate sampling rate, knowledge of the signal is critical. We are most interested in the peak force and loading rate leading up to it. Preliminary testing showed that this rise time is fastest for the shoulder taps and can happen as quickly as a few milliseconds. Conservatively assuming this rise occurs over 1 millisecond, a sampling rate of 50 kHz leads to 50 samples in this critical measurement period. A number deemed sufficient for our purposes and within the capabilities of the measurement system.

The "tap-o-meter" was initially developed using parts in stock at the Norwegian Water Resources and Energy Directorate (NVE). Early testing suggested that a ~500 N load cell which NVE had in stock would be capable of accurately recording the impact force from taps. Based on data collected prior to those showcased in this paper, it became evident that the impact forces from some participants plateaued around 600 N on their shoulder taps. This level surpassed the recommended operating range

of the load cell but stayed within the ultimate overload capacity (~1,500 N). We pinpointed the problem to the amplifier, which was reaching its saturation point.

We considered the amplifier properties to avoid two potential issues. Setting it too high would mean losing detail in measuring light wrist taps due to an increased background noise. On the other hand, setting it too low would make it impossible to measure the strongest impact forces.

To address this, we developed a new adjustable amplifier that we tuned to a range from 5 to 1,000 N. This calibration aimed to balance the ability to detect high-impact forces while maintaining a low background noise for measuring the force of lighter taps. The defined range stayed safely below the load cell's ultimate overload threshold of 1,225 N. Despite the new adjustment with the amplifier's upper limit set to 1,000 N, saturation still occurred in rare instances: once during elbow-level taps (representing 0.03% of such taps) and 75 times for shoulder-level taps (2.63% of such taps).

## 2.2 Data collection process

Data collection was conducted at events in Norway, Switzerland, Austria, USA, and Canada. In Norway, data was collected from avalanche forecasters and mountain guides. In Switzerland, data was collected at the European Avalanche Warning Service (EAWS) general assembly. Canadian and Austrian events only included avalanche forecasters. Events in the USA contained a mix of avalanche workshop participants and avalanche forecasters. A total of 286 individuals (232 males and 54 females) contributed to the study. A detailed table of the number of samples, event, and date can be found in Appendix-2. We did not provide any specific instructions on how to conduct the ECT other than that we asked participants to tap as they would do in the field. We provided a wide range of gloves with different thicknesses, but it was up to the participants themselves to select which glove, or whether to use a glove at all.

We made the setup as similar as possible by using three identical "tap-o-meter" devices. All "tap-o-meters" were firmly attached to a wooden CT (30 x 30 x 85 cm) or ECT (30 x 90 x 85 cm) column (Fig. 1). By using a fixed height, we acquired data with a consistent sampling method but are not able to adjust for changes in simulated snowpack thickness. Furthermore, participants were given the choice to use different types of gloves depending on their preferences. The intent was that all participants should be able to conduct the test like they would do in the field. However, we left the shovel handle off as early tests during the development showed that even gentle touches are picked up with our sensitive load cell.

### 2.2.1 Survey

For each participant, we asked them to fill out a survey where they noted their country of residency, avalanche climate, height, weight and gender. The information from the survey was collected to answer the following research questions:
1. Does height, weight, and/or gender affect tapping force?

2. Do people tap differently across avalanche climates?

3. Are there regional differences between Scandinavia, Alps and North America?

## 2.3 Data processing

The raw voltage data are processed using python to identify the individual taps. After the taps are identified, two metrics are pulled from each one: maximum force (newtons, N) and loading rate (N/s). Other quantities such as impact duration, rise time, and stress were considered but not chosen. Impact duration was not used because the measurements frequently contained long, oscillatory tails that are artifacts of the load cell rebounding and vibrating – a phenomenon expected to be less present during an actual field test. Rise time is calculated as an intermediary step to loading rate. However, loading rate was chosen because

snow's response has been shown to depend on its rate of deformation (Shapiro et al., 1997, Reiweger et al., 2015, Verplanck and Adams, 2024). Lastly, our measurements are presented as forces (N) rather than stresses (kPa) because presenting it as a stress would rely on an assumption of cross-sectional area.

The recorded time and voltage are imported as NumPy arrays (Harris et al., 2020). The voltage values are zeroed by subtracting

the entire array's mean from each data point. Then, voltage is converted to newtons by scaling according to the calibration. Scipy's (Virtanen et al., 2020) peak finding algorithm, scipy.signal.find_peaks, is implemented to determine when the taps occur by comparing neighboring values. The peak finding algorithm is driven with two parameters: a 25 N minimum peak magnitude and 0.4 seconds minimum time between peaks. These criteria are chosen by iteratively trying different values and viewing the results. This peak finding method is used as a first pass through the data and is later refined with a more manual

process. See Figure 2 for an example of tap data with the peaks algorithmically identified.

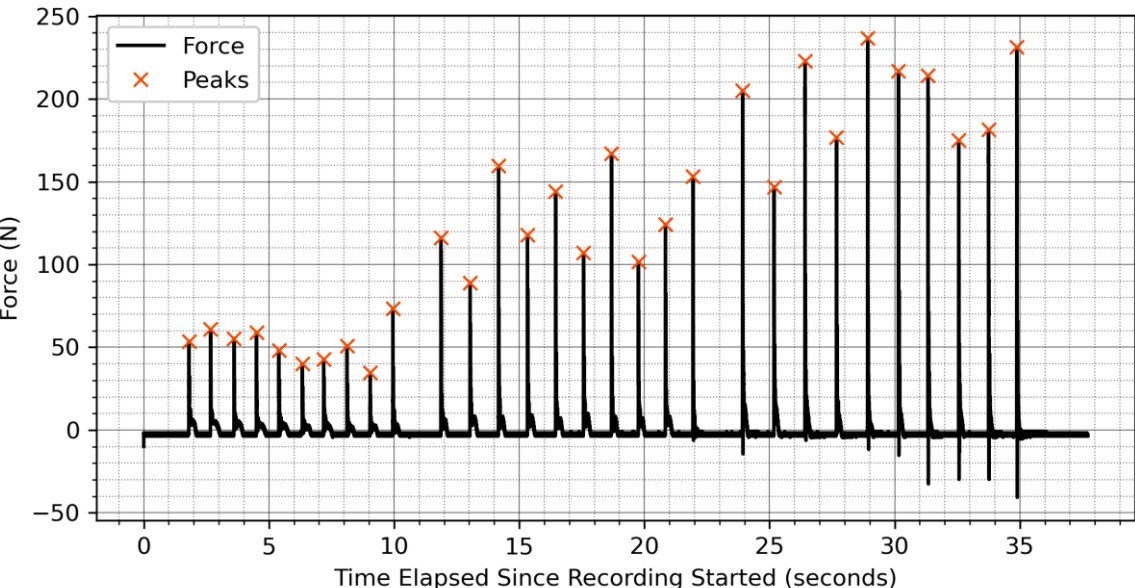

Recorded Time Series of Force with Identified Peaks

**Figure 2: An example of identifying taps using SciPy's peak finding algorithm with 25 N minimum peak magnitude and a minimum of 0.4 seconds between peaks. Using these parameters, the algorithm correctly identified all peaks as it did in 262/286 cases. Manual adjustments to the algorithm's parameters were used in the remaining 24 cases to identify peaks.**

After the peaks are found the individual taps are defined as 70 ms prior to and 40 ms after the peak. These values are chosen to allow for enough time surrounding the peak to determine tap metrics. Each tap array is then re-zeroed by subtracting the mean of the first 0.2 ms of that specific tap. This re-zeroing process is implemented because subtle shifts in the baseline recording are occasionally apparent, particularly during the taps hinging from the wrist if the tapper kept contact with the shovel blade throughout these taps. The two metrics, maximum force and loading rate, are ascertained from each tap array. Maximum force, $F_{peak}$, is simply the maximum value in the re-zeroed array. The loading rate, $r$, is defined as a linear interpolation, Eq. (2), between the maximum force, $F_{peak}$, and a threshold value greater than typical noise, $\lambda$. In our measurements, a $\lambda$ of 15 N was deemed appropriate. The difference in force is divided by the rise time, $\Delta t$, to determine the loading rate. The rise time is the difference in time between the peak force and the initial threshold crossing.

$$r = \frac{(F_{peak} - \lambda)}{\Delta t} \tag{2}$$

After this automated process is applied to all 286 tap recordings, a manual quality control process is done. This process entails viewing the taps for each recording (Fig. 3), flagging misidentified taps, and classifying which taps are hinging from the wrist, elbow and shoulder. This manual process determined that 262/286 recordings were correctly processed with the first-pass algorithm. The remaining 24 recordings were reprocessed by changing the parameters for SciPy's peak finding algorithm. The

changes to peak-finding parameters involved reducing the time between peaks or minimum magnitude until all the clear taps are identified. In some cases, the metrics were not calculated accurately because there was a spike of noise that was close enough in time to the tap signal. In these cases, the individual taps were not included in the analyzed data set.

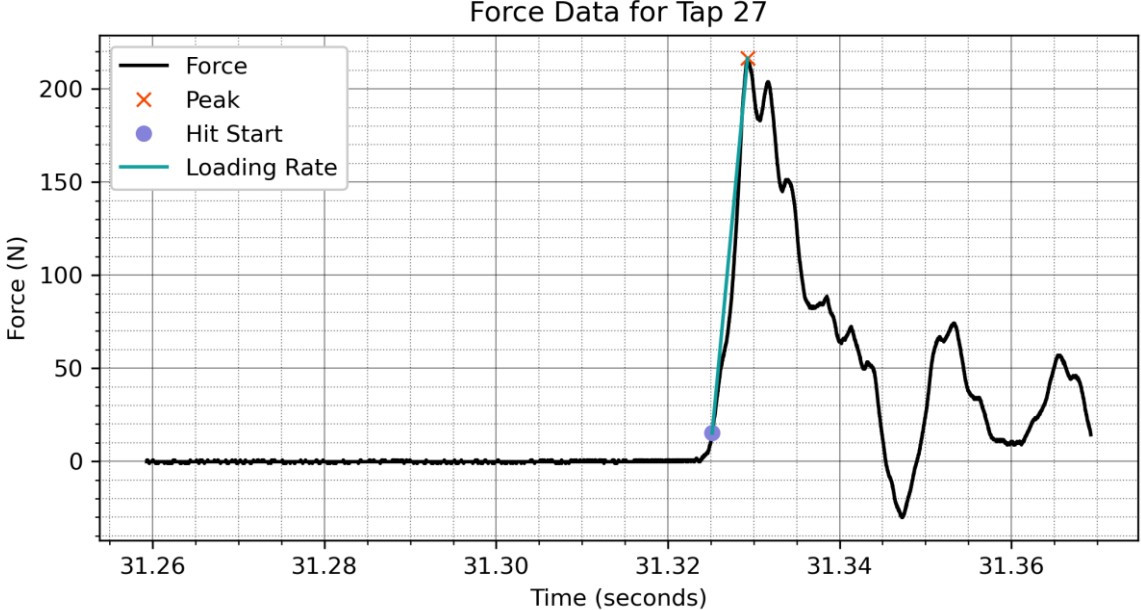

**Figure 3: An example of the data processing procedure implemented on a shoulder tap. This procedure acquires two metrics for**
**each tap: peak force (N) and loading rate (N/s).**

## 2.4 Statistical analysis

We tested height, weight, gender, and geographic region to understand the underlying factors influencing hand-tap loading using ordinary least squares (OLS) regression models. The peak force was the dependent variable in these models. To compare hand-tap loading at different loading steps, we conducted a one-way ANOVA. This analysis assessed whether the mean impact
forces were statistically different during wrist, elbow, and shoulder taps. ANOVA, or Analysis of Variance, compares the means of three or more groups to determine if at least one group's mean is significantly different from the others (Fisher, 1970). All analyses were considered statistically significant at p-values below 0.05.

## 2.5 Idealization of taps as Gaussian functions

Both the peak force, $F_{peak}$, and loading rate, $r$, are used to idealize the impact curves. First, consider the equation describing
a Gaussian function of force, $F$, as a function of time, $t$.

$$F(t) = F_{peak}e^{-\frac{1}{2}\left(\frac{t-t_{peak}}{\sigma}\right)^2}, \tag{3}$$

Where $F_{peak}$ is the peak force and $t_{peak}$ is the time at which the peak force occurs. The duration of the force curve is governed by $\sigma$, the standard deviation if the Gaussian function were to be describing a normal distribution. Since 99.7% of the curve's magnitude occurs during $6\sigma$, the duration of impact is defined $6\sigma$ in our study. Thus, the rise to peak force occurs over approximately $3\sigma$, leading to the following relationship to calculate the loading rate, $r$.

$$r \approx \frac{F_{peak}}{3\sigma}, \tag{4}$$

This is an approximation rather than equality because it assumes a linear rise, rather than the non-linear Gaussian shape. However, since loading rate and peak force are the two metrics ascertained from the measured data, this approximation provides a convenient way to idealize the measured force curves. Rearranging the approximation yields

$$\sigma \approx \frac{F_{peak}}{3r}, \tag{5}$$

And substituting this relationship for $\sigma$ in Eq. (3) yields the Gaussian approximation used to idealize the measured force-time curves.

$$F(t) \approx F_{peak}e^{-\frac{1}{2}\left(\frac{3r(t-t_{peak})}{F_{peak}}\right)^2}, \tag{6}$$

## 3. Results

### 3.1 Peak force and loading rate

The data set consists of 2,837 wrist taps, 2,839 elbow taps, and 2,846 shoulder taps across 286 individuals. Outliers are excluded using 1.5 times the interquartile range (IQR) method, which is a widely recognized and accepted standard in statistical analysis (Tukey, 1977). Saturation occurred in rare instances due to a limitation with the amplifier in the "tap-o-meter". See Table 2 for more information.

**Table 2: Number of taps, outliers and saturation taps for peak force and loading rate.**

|  | Peak Force | | | Loading Rate | | |
|---|---|---|---|---|---|---|
|  | Wrist | Elbow | Shoulder | Wrist | Elbow | Shoulder |
| No. of taps | 2,837 | 2,839 | 2,846 | 2,837 | 2,839 | 2,846 |
| No. of outlier taps | 119 (4.2%) | 93 (3.3%) | 123 (4.3%) | 149 (5.2%) | 108 (3.8%) | 205 (7.2%) |
| No. of saturation taps | 0 (0.0%) | 1 (0.0%) | 75 (2.6%) | 0 (0.0%) | 0 (0.0%) | 0 (0.0%) |

In Table 3, we provide some descriptive statistics of peak force and loading rate. The median peak force approximately doubles from one loading step to the next at 79 N, 185 N and 373 N respectively. The standard deviation is also roughly half of the mean peak force for each loading step, showing that the variability in loading increases proportionally with increasing peak force. The loading rate, and its standard deviation, increases with each load step. The loading rate is positively correlated with peak force ($R^2 = 0.70$).

**Table 3: Descriptive statistics of peak force and loading rate (outliers removed using 1.5 * IQR).**

| | Peak Force (N) | | | Loading Rate (N/s) | | |
|---|---|---|---|---|---|---|
| | Wrist | Elbow | Shoulder | Wrist | Elbow | Shoulder |
| Mean | 79 | 185 | 373 | 8,819 | 28,836 | 66,088 |
| Standard deviation | 39 | 82 | 172 | 6,745 | 17,362 | 41,951 |
| Min | 8 | 34 | 45 | 118 | 149 | 2,316 |
| 25th percentile | 50 | 123 | 239 | 3,449 | 15,107 | 37,128 |
| Median | 73 | 173 | 343 | 6,842 | 25,068 | 61,553 |
| 75th percentile | 101 | 237 | 481 | 12,763 | 39,830 | 90,676 |
| Max | 190 | 426 | 893 | 30,145 | 81,619 | 195,812 |

We observed different mean and median values for each loading step, and if we consider the interquartile range, which represents the data between the 25th and 75th percentile, there is nearly no overlap between loading steps. Doing a one-way ANOVA, we get a p-value lower than 0.01, indicating that the three loading steps are statistically different from each other, mirroring the findings of Sedon (2021) and Griesser et al. (2023).

In Figure 4, the distribution of peak forces across different tap numbers is graphically represented for three tapping levels. While the median forces across each loading step remain relatively consistent, there is a large spread across all loading steps. Collectively, this figure emphasizes the inherent differences in peak forces across the three tapping levels and underscores the variability present within each level across different tap numbers.

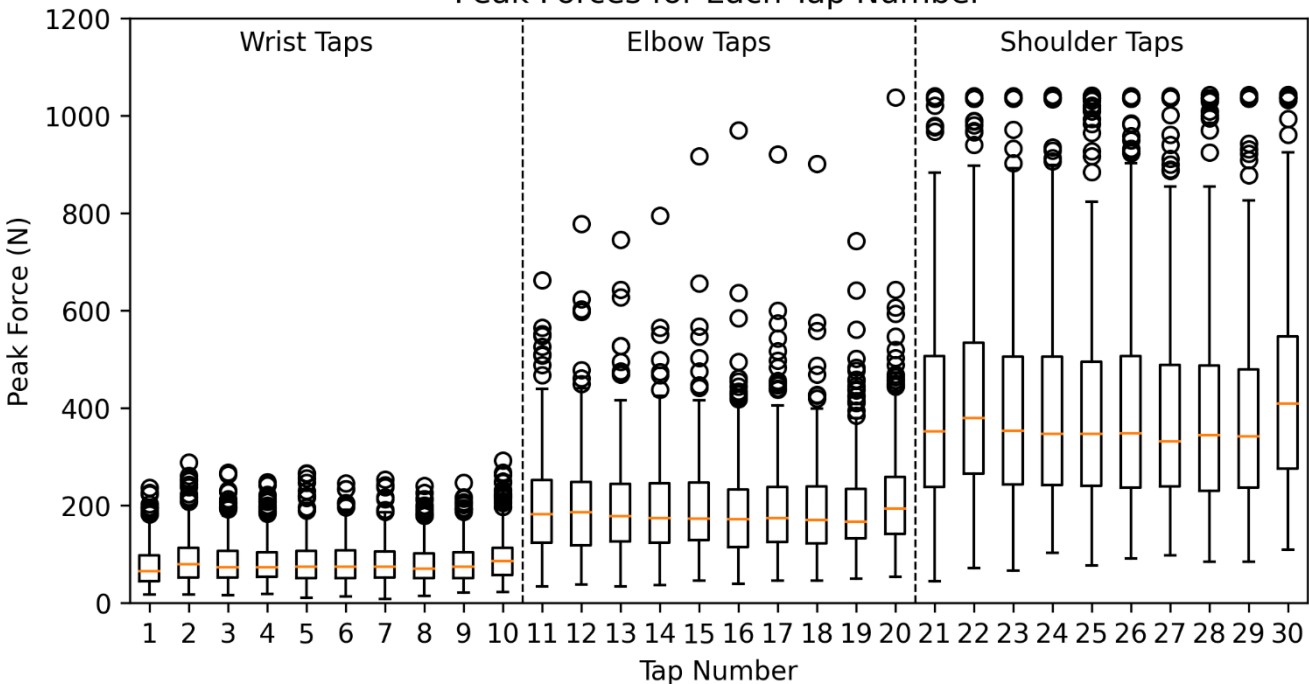

Figure 4: A visualization of the magnitude and variability in peak impact force from the 286 participants from tap 1 to 30. A box plot for each tap number displays the minimum, first quartile, median, third quartile, and maximum values. Outliers are shown using circular symbols. The load cell reaches saturation at 1,000 N, a threshold which was reached in one elbow tap and 75 shoulder taps.

To showcase the overlap between loading steps, we have made a confusion matrix based on a tapping norm. The IQR for wrist, elbow and shoulder is, respectively, 50-101 N, 123-237 N and 239-481 N. We have selected the value between the highest IQR value in one loading step and lowest IQR in the next to define the tapping norms between loading steps. For example, the upper bound wrist norm is 112 N which lies halfway between 101 N and 123 N. The lower bound for the wrist norm is the 25$^{th}$ percentile threshold, and the upper bound for the shoulder norm is the 75$^{th}$ percentile threshold. Using these values, we can make a confusion matrix to highlight how many hand taps that are within each interval (Table 4). From this, we can see, for example, that 17.79% of elbow taps are within the wrist tapping norm and 25.75% are within the shoulder norm.

Table 4: A confusion matrix based on the tapping norm. The table highlights how large of a proportion of the peak forces for wrist, elbow and shoulder taps fall within each tapping norm.

| | < Wrist (< 50 N) | Wrist (50-112 N) | Elbow (112-238 N) | Shoulder (238-481 N) | > Shoulder (> 481 N) |
|---|---|---|---|---|---|

| | | | | | | | | | | | | | | | |
|---|---|---|---|---|---|---|---|---|---|---|---|---|---|---|---|
| **Wrist** | 23.48% | | 53.79% | | 21.71% | | 1.02% | | 0.00% | | | | | | |
| **Elbow** | 0.92% | | 17.79% | | 53.82% | | 25.75% | | 1.73% | | | | | | |
| **Shoulder** | 0.04% | | 1.30% | | 22.24% | | 48.70% | | 27.72% | | | | | | |

## 3.2 Explanatory factors' correlation with peak force

The three panels in Table 5 contain the results for the different loading steps. Panel 1 shows the result for taps from the wrist, panel 2 for taps from the elbow, and panel 3 for taps from the shoulder. We have estimated five models for each type of tap to evaluate the role of weight (model I), height (model II), and gender (model III), respectively. Models IV and V adds a control for gender to the height and weight variables.

Table 5. Results from OLS regression. Standard errors in parentheses.

| | ln (wrist) | | | | | ln(elbow) | | | | | ln(shoulder) | | | | |
|---|---|---|---|---|---|---|---|---|---|---|---|---|---|---|---|
| | I | II | III | IV | V | I | II | III | IV | V | I | II | III | IV | V |
| Weight | 0.008** | | | 0.005+ | | 0.006** | | | 0.002 | | 0.006* | | | 0.001 | |
| | (0.002) | | | (0.003) | | (0.002) | | | (0.003) | | (0.002) | | | (0.003) | |
| Height | | 0.011** | | | 0.008+ | | 0.010** | | | 0.004 | | 0.008* | | | 0.000 |
| | | (0.004) | | | (0.004) | | (0.003) | | | (0.004) | | (0.003) | | | (0.004) |
| Female | | | -0.221** | -0.137 | -0.121 | | | -0.256** | -0.218** | -0.199* | | | -0.269** | -0.256** | -0.266** |
| | | | (0.076) | (0.089) | (0.095) | | | (0.070) | (0.080) | (0.088) | | | (0.065) | (0.089) | (0.091) |
| Region (reference is European Alps) | | | | | | | | | | | | | | | |
| North America | 0.085 | 0.106 | 0.120+ | 0.100 | 0.113+ | 0.040 | 0.058 | 0.074 | 0.065 | 0.070 | 0.095 | 0.111+ | 0.126* | 0.123+ | 0.126* |
| | (0.068) | (0.068) | (0.067) | (0.068) | (0.067) | (0.059) | (0.059) | (0.058) | (0.057) | (0.057) | (0.064) | (0.064) | (0.063) | (0.063) | (0.063) |
| Scandinavia | -0.041 | -0.031 | -0.002 | -0.023 | -0.018 | -0.174* | -0.167* | -0.136* | -0.146* | -0.146* | -0.084 | -0.076 | -0.047 | -0.051 | -0.048 |
| | (0.080) | (0.079) | (0.081) | (0.080) | (0.079) | (0.068) | (0.067) | (0.067) | (0.068) | (0.066) | (0.070) | (0.068) | (0.068) | (0.071) | (0.069) |
| Constant | 3.718** | 2.335** | 4.304** | 3.933** | 2.955** | 4.717** | 3.444** | 5.225** | 5.061** | 4.470** | 5.426** | 4.477** | 5.887** | 5.829** | 5.847** |
| | (0.187) | (0.637) | (0.052) | (0.222) | (0.801) | (0.179) | (0.588) | (0.038) | (0.209) | (0.742) | (0.183) | (0.587) | (0.042) | (0.255) | (0.801) |
| N | 286 | 286 | 286 | 286 | 286 | 286 | 286 | 286.000 | 286 | 286 | 286 | 286 | 286 | 286 | 286 |
| F-value | 4.809 | 4.292 | 4.385 | 4.007 | 3.782 | 5.294 | 6.406 | 7.799 | 5.835 | 6.177 | 3.649 | 3.800 | 8.722 | 6.517 | 6.519 |
| R2-adjusted | 0.032 | 0.035 | 0.031 | 0.036 | 0.037 | 0.051 | 0.057 | 0.072 | 0.070 | 0.072 | 0.033 | 0.032 | 0.062 | 0.058 | 0.058 |
| AIC | 424.658 | 423.625 | 424.960 | 424.503 | 424.046 | 359.685 | 357.627 | 353.313 | 354.705 | 354.146 | 388.486 | 388.669 | 379.849 | 381.780 | 381.846 |

+ $p<0.1$, * $p<0.05$, ** $p<0.01$

Overall, the models explain very little of the variance in peak tap force (between 3.1% and 7.2%). In other words, over 90% of peak tap force variance is explained by factors other than height, weight, gender, and geographical region. While we do find a significant positive correlation between peak tap force and both height and weight, the effects are very small. An increase in weight by one kilo is associated with an increase in peak force by 0.6% to 0.8% in our sample. The effect of height is slightly larger, but still very small. An increase by one centimeter is associated with an increase in peak force by about 1%. In addition,

in the models for taps from the elbow and shoulder, the effects of height and weight drop below 10% significance when we control for gender. The models for elbow and shoulder taps further suggest that gender is a more important explanatory factor than height and weight, as can be seen by the relatively larger R2-adjusted values for models where gender is included. This

result does not hold for wrist taps, where gender is an equally poor (if not poorer) predictor of peak tap force as weight and height. In general, our results suggest that women's peak force is about 20% less than men's peak force.

### 3.3 Gaussian function idealization

Using the median metrics along with their 25th and 75th percentiles (Table 3), the force curves idealized as Gaussians are shown in Figure 5.

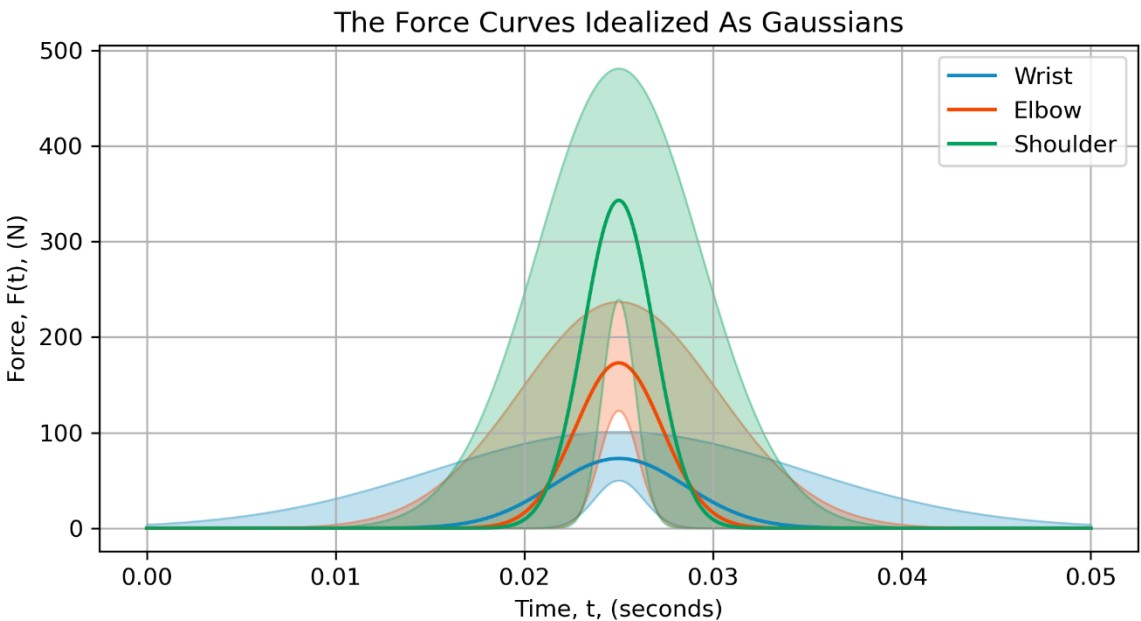

**Figure 5: An idealization of the taps as Gaussian functions. The center lines are from the median metrics and the shading is generated from the 25th and 75th percentiles.**

By idealizing these tap curves as Gaussians, their respective linear impulses can be compared by calculating the area under curve (Hibbeler, 2010). Using NumPy's implementation of the trapezoidal rule (Harris et al., 2020), the median wrist, elbow, and shoulder tap impulses are 0.65, 1.00, and 1.60 N*s, respectively. We estimate the median loading duration ($6\sigma$, section 2.5) of the impact curve to be 21 ms for the wrist, 14 ms for the elbow, and 11 ms for the shoulder.

### 4. Discussion

Using the data from the "tap-o-meter", we can provide insight into the impact forces of hand taps and the variability between participants. We believe the quantification of the magnitudes and variabilities associated with hand-tap loading will assist with our understanding and interpretation of the ECT and CT.

## 4.1 Comparison of peak applied force to other studies

If we compare the results from our study with the ones from Sedon (2021) and Griesser et al. (2023), we find surprisingly large discrepancies when comparing the measured mean values (Table 6). It is unlikely that participants from New Zealand (Sedon, 2021) tap half as hard as Griesser et al. (2023) observed or one-third of what we observe in our sample from Scandinavia, Europe, and North America. Griesser et al. (2023) recognize that they are not able to accurately measure peak force values due to their lower sampling rate but that the relative differences are systematic when comparing the mean values from wrist, elbow,

and shoulder with data from our study. We have measured the 62 participants from Griesser et al. (2023) in parallel with our own measurement device, and the measurements are very similar to the rest of our samples. This comparison suggests that the differences are likely due to the difference in sampling rate.

Table 6: A comparison of mean peak force values for wrist, elbow, and shoulder from relevant studies.

| Reference | Wrist (mean) | Elbow (mean) | Shoulder (mean) | Sampling rate | Samples |
|---|---|---|---|---|---|
| Sedon (2021)[1] | 24 N | 62 N | 136 N | Unknown | 69 |
| Griesser et al. (2023) | 41 N | 97 N | 185 N | 100 Hz | 62 |
| This study | 79 N | 185 N | 373 N | 50 kHz | 286 |

[1] Sedon (2021) uses the maximum value from each loading step to calculate the mean between participants.

At a sampling rate of 100 Hz, we would only measure the impact force every 10 ms, making it unlikely to capture the peak force value accurately. The discrepancies in sampling rates make for an invalid comparison of peak force values between the studies. However, the relative difference between wrist, elbow, and shoulder is almost identical for all studies. All three studies

have an approximately doubling in peak impact force from wrist to elbow to shoulder.

## 4.2 Body characteristics, gender, and region

Sedon (2021) did not investigate whether there were differences due to weight, height, gender, or geographical region. Griesser et al. (2023) investigated shoulder height and found that participants with greater shoulder height had higher impact forces. They also mention that they found statistically significant correlations when comparing against height and weight, but no p-

values are provided. Our main finding from the survey data is that only gender has a statistically significant relationship with peak force. Body features (weight and height) are also correlated with peak tap force, but when included in a multivariate analysis with gender, they disappear. We believe the correlation found by Griesser et al. (2023) for body features is likely due to men being, in general, taller, and heavier.

Given the variations in observational guidelines for the ECT, we hypothesized that measuring differences among participants from the Alps, Scandinavia, and North America would be feasible. Despite this expectation, we observed no regional variations

in peak tapping force. The lack of significant findings might be attributed to our limited predictive capability from the small sample size in a statistical context (n=286), or that there are no differences to be found.

### 4.3 Variability in tapping force – implications for stability interpretations

It is widely agreed that whether a crack propagates across the entire column or not is the key discriminator between unstable and stable slopes (Techel et al., 2020). However, both Winkler and Schweizer (2009) and Techel et al. (2020) show that the number of taps provides additional information, allowing a more refined distinction between results related to stable and unstable conditions. Techel et al. (2020) found the optimal threshold between ECTP20 and ECTP22, which aligns with the ECTP21 threshold suggested by Winkler and Schweizer (2009). Moving away from a binary classification came at the cost of

introducing intermediate stability classes (Techel et al., 2020).

These new intermediate stability class definitions rely heavily on the tap number when failure occurs. Variability in the applied force-time curves likely leads to variability in test results, particularly regarding the number of taps required to induce weak layer failure. It is important to emphasize that no tests offer a definitive "go/no go" result. With accuracies of around 80%,

these tests are not reliable enough to be the main factor in our slope scale decision-making (Birkeland et al., 2023).

We found the three loading steps to have statistically different IQRs; this aligns with the results from Griesser et al. (2023), which highlight this as a positive outcome and that the CT and ECT hand-tap procedure is somewhat reliable. Despite the statistical differences in each loading step, we question the application of average results to individual cases. The main

difference in our argument lies in relying solely on mean statistics to develop tapping norms used by individuals. For example, from Table 4, we can see that 17.79% and 25.75% of elbow taps have a peak force value that falls within the tapping norms for wrist and shoulder taps, respectively. This implies that 43.54% (17.79% + 25.75%) of elbow taps would be misclassified as taps hinging from the wrist or shoulder. Assuming peak applied force influences test results, then this misclassification of loading steps will lead to a misclassification of test results. Because stability tests results aid in an individual's decision-making

process, a misclassification of test results could lead to dangerous consequences in real-world applications.

### 4.4 Idealization of taps as Gaussian functions

The Gaussian function is often used in wave propagation problems because it represents a smooth, continuous pulse of disturbance (Langtangen & Linge, 2017). The measured shape of force-time curves is not a perfect Gaussian (Fig. 3), particularly after the peak force has been reached. The noisy, oscillatory decay following the peak is attributed, in part, to the

instrumentation. Despite these imperfections, we intend to use this idealization as a steppingstone towards mathematical modelling efforts. In addition to providing this steppingstone, the idealization shown in Figure 5 provides a visualization of peak force, loading rate, impact duration, and variability associated with these quantities. The taps from the shoulder are generally a sharper pulse (i.e. shorter duration, higher peak force) than a wrist tap. Despite the impact duration decreasing

with increasing load step, there is an increase in linear impulse. The linear impulse is equated to the change in linear momentum
of the system (Hibbeler, 2010). Thus, the increase in snow's momentum from a hand tap is expected to be larger for higher load steps despite the shorter duration of impacts. The Gaussian idealization provided a convenient method of comparing linear impulses from the tap data whereas direct numeric integration of the load cell data would be inaccurate due to the long, oscillatory tails.

## 4.5 Implications for avalanche practitioners

Given the variability in tapping demonstrated in this study, we propose two considerations to improve the ECT standards. The two ideas outlined below are intended to be a foundation for further discussion in the broader avalanche community.

### 4.5.1 Reduce tapping variability through the use of training and/or tools.

The large variability in impact force between individual participants highlights the need for standardization. This could be done by creating a better definition of how the test should be conducted in terms of technique and tapping force. When
interpreting the descriptive definitions from each loading step, it is impossible to infer which impact forces should be used as a baseline for each loading step. For example, the Norwegian description (Norwegian Water Resources and Energy Directorate, 2022) using the arm's weight would depend on the weight of each participant's arm. Furthermore, using Canada as an example, there is no description of how hard each tap should be other than that it should not hurt at shoulder level (Canadian Avalanche Association, 2016). However, this would depend on the participant's pain tolerance, snow properties (dampening) and the
participant's glove thickness.

The community will need to agree on what the ideal impact force-time curves are. The impact forces presented in this paper could be used as a baseline for future clarifications if a "wisdom of crowds" impact force definition is employed (see Surowiecki, 2005 for an introduction to the concept of "wisdom of crowds"). An alternative to the "wisdom of the crowds" is
a selection of experts could choose to define the appropriate windows and thresholds.

With these windows defined, a training device could be developed that measures the impact force and reports back to the participants whether they are within the correct window at each hand loading step. If a training device is considered to be the best solution to reduce interpersonal variability, we believe this paper provides sufficient information to build such a training
device. Such devices already exist for CPR training and provides real-time measured feedback on compression rate (cpm), depth (mm), release (g), compressions count, and inactivity time during CPR, while also enabling responders to self-evaluate their performance with event statistics on the spot (Laerdal, 2023).

Another solution could be to develop a tool that ensures consistent impact force, like the stuffblock test (Johnson and Birkeland,
1998). The test involves filling a nylon sack with 4.5 kg of snow and dropping it in increments of 10 cm. However, this test

type of loading has its challenges. The peak force and loading rate are coupled and depend on the object's mass, the drop height, and the materials that are in contact during impact. Not only mass and height would need to be recommended, but also materials and possible use of cushion-like material to recreate both peak force and loading rate of hand taps. Verplanck and Adams (2024) attempted to match the impact curves of hand taps using an acetal mass, foam cushion, and aluminum plate.
However, they attempted to match their own hand taps, not the averages presented in our study.

### 4.5.2 Revisiting the stability interpretation of CT and ECT

Our second proposition comes from the implication of defining predictor thresholds based on impact forces from a large database of ECTs. The concern is that the large variability in hand-tap loading makes these average-based thresholds relatively weak. The thresholds make sense when analyzing large amounts of data (e.g. in the context of avalanche forecasting) but not
when applying the average results to individual cases. We should therefore evaluate whether the importance of the number of taps outweighs the risk of misinterpreting the test result.

One thought example could be whether it is appropriate to interpret ECTP20 (intermediate stability) compared to ECTP24 (unstable) in individual cases (Winkler et al. 2009), given the large discrepancies in impact force. There is also precedent for
adopting a more straightforward approach in interpreting ECT results at the expense of leaving potentially relevant information out, as when shear quality and fracture characteristics were removed from the ECT (Simenhois et al., 2018). In this approach, we would consider the test result to be unstable if crack propagation occurs, and stable otherwise. When using the more simple, binary approach, the impact force becomes less important, and the large variation is less of a problem.

### 4.6. Limitations

### 4.6.1 The "tap-o-meter"

While our study has made strides in accurately observing the force-time curves from hand taps, there are still areas that require further exploration. For instance, tap force measurements greater than 490 N may not be as accurate force measurements below 490 N because 0-490 N is the recommended load cell range. Also, our calibration assumes the load cell responds similarly to dynamic loads as static loads and eccentric loads as centered loads. These potential inaccuracies in the measurement technique
likely contribute to the range and variability of force measured in this study. Future studies should therefore include a load cell with a higher range (e.g. 2000 N), load cells designed for impacts (e.g. piezo-resistive), and a fixture to ensure centered loading. By doing so, we can enhance the precision, accuracy, and reliability of our measurements, leading to more robust and accurate findings. Despite these potential measurement inaccuracies, our study utilized a sampling rate (50 kHz) appropriate for capturing the entirety of the impact curve. This is an improvement over similar studies that used a sampling rate of 100 Hz.
(Griesser et al. 2023) and 105 Hz (Thumlert and Jamieson, 2015). Sedon (2021) do not provide any sampling rate for their study.

### 4.6.2 Data collection

Initially, our idea was to have a representative group of participants with different levels of training. However, after the first data collection event, we realized that most novices did not know how to do the test, and it was difficult to get a representative sample from less experienced participants. Each participant was asked to fill out a survey. In retrospect, an estimate of how many ECTs each participant does in a season would be of interest. Most participants noted that they do it regularly at work, recreation, or both, but we do not have an idea of how frequently they conduct ECTs.

Furthermore, systematic notes about the tapping technique would also be of interest. A qualitative remark is that many of the participants infrequently use their fingertips on wrist taps as in the standards (American Avalanche Association, 2022; Canadian Avalanche Association, 2016). There was also a large variability in impact forces because of different techniques such as using the weight of the arm versus a shoulder tap so hard that it hurts the hand. In some cases, participants placed a glove on the shove to soften the blow. We also observed that some participants increased their impact force during the ten taps within each level, but we do not see this in our overall data (Figure 4).

### 4.7 Future work

During data collection, we asked participants if they regularly conduct CTs or ECTs for work, recreation or both. Participants were also asked to self-evaluate their avalanche assessment level on a scale from 1 to 6, following the definitions from the CARE-panel study (Hetland & Mannberg, 2023). Our hypothesis was that more experienced participants, particularly those frequently performing stability tests, would be more consistent within each loading step. However, the study's shift in focus towards more experienced individuals (see Section 4.1.2) meant that we lacked a suitable reference group for comparison. For future studies, a more effective approach might involve quantifying the frequency of CTs or ECTs performed by each participant per season. This method could provide a more nuanced understanding of the relationship between the quantitative experience and tapping consistency.

Snow's response to impact forces remains an active research topic and is out of the scope of this study. However, variability in magnitude and duration of applied force will result in variability of the stress state within the snow which may lead to variability in test results. For more on this topic, we refer the reader to studies by Napadensky (1964), Wakahama & Sato (1977), Johnson et al. (1993), Schweizer et al. (1995), van Herwijnen & Birkeland (2014), Thumlert & Jamieson (2015), Griesser et al. (2023), and Verplanck and Adams (2024). Quantifying how variability in the applied force may lead to different ECT results would be a useful extension of our work presented here.

## 5. Conclusion

In this study, we developed a device that can accurately measure force-time curves from the hand-tap loading method. We emphasize the importance of sampling rate to accurately measure these curves, leading us to implement a sampling rate of 50 kHz – a recommended value for future studies as well. The dataset collected is the largest one to date (286 participants, 8522 taps), including data from Scandinavia, the Alps, and North America. From these data, we quantified peak force and loading rate for each tap, both of which increased for each loading step (i.e. wrist, elbow, shoulder). There is nearly no overlap in peak force from the 25th to 75th percentile between loading steps. Yet there is significant overlap in the outer quartiles with examples of some wrist taps with as high of peak force as others' shoulder taps. An exploration into defining tapping norms based on the inner quartile range of peak force is presented. However, due to the overlapping outer quartiles, almost half of elbow taps would be misclassified as taps hinging from the wrist or shoulder. Assuming peak applied force influences stability test results, then this misclassification of loading steps will lead to a misclassification of stability test results.

Using the observed peak forces and loading rates, the force-time curves are idealized as Gaussian functions. This idealization provides a convenient steppingstone for future mathematical modeling efforts of stability tests like the Compression Test and Extended Column Test.

We investigated whether the differences in weight, height, gender, and/or geographical region influence peak force using multivariate statistical models. Overall, these variables explain very little of the variance in peak tap force, with over 90% of the variance attributed to factors other than height, weight, gender, and geographical region. Our results indicate that gender is the only statistically significant explanatory variable, with women's peak force being approximately 20% less than men's peak force.

Our results provide an answer to the question of "How hard *do* avalanche practitioners tap?" but not necessarily "How hard *should* avalanche practitioners tap?". We recommend that our data be used to facilitate discussions related to updating guidelines for the hand-tap loading method, possibly of including thresholds of peak force and loading rate for each loading step. Given the variability in tapping demonstrated in this study, we propose two considerations to improved standards: (1) reduce tapping variability through the use of training and/or tools and (2) evaluate whether the importance of the number of taps outweighs the risk of misinterpreting the stability test results.

### Data availability

The data needed to replicate the study is available in our Open Science Framework repository (Toft et al., 2023).

## Author contributions

The study was conceptualized by HT, SV, and ML. HT developed and built the three "tap-o-meters". All authors actively participated in data collection at various events. SV, with HT's assistance, conducted the data pre-processing. HT led the analysis on trends and variability among participants, incorporating insights from SV and ML. The conceptualization of taps as Gaussian functions was primarily driven by SV, with inputs from HT and ML. All authors were actively involved in the preparation, editing, and review of the original draft.

## Acknowledgements

We would like to acknowledge Knut Møen for his technical contributions to the development of the "tap-o-meter" and for his creative input in naming the device. Furthermore, Andrea Mannberg for her statistical expertise, Christoph Mitterer and Scott Savage for assistance in data collection – with additional thanks to Scott for facilitating us while working on this in Idaho. Jordy Hendrikx for connecting the authors, a collaboration born out of the realization that we were doing similar work. Thank you to all the study participants as well.

## Competing interests

The authors declare that they have no conflict of interest.

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

**Appendix**

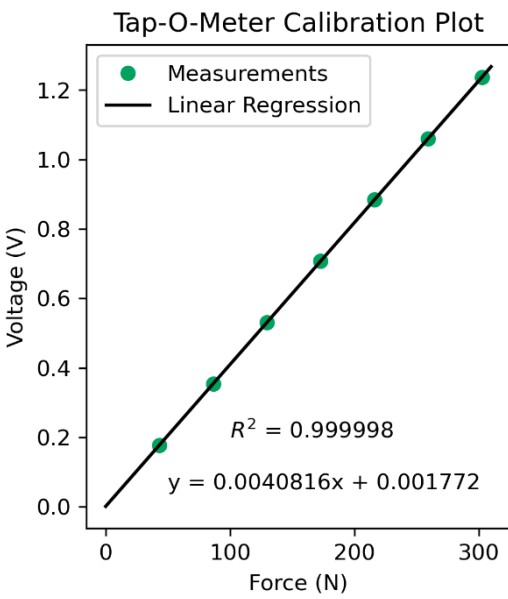

670

Appendix-1: The "tap-o-meter" was calibrated using known weights ranging from ~50 to 300 N.

| Event | Date | Samples |
| --- | --- | --- |
| European Avalanche Warning Services General Assembly | 15.06.2022 | 62 |
| Montana State University Snow and Avalanche Workshop | 26.10.2022 | 25 |
| Norwegian Avalanche Observer Workshop | 08.11.2022 | 46 |
| UIAGM General Assembly Norway | 12.11.2022 | 27 |
| Friends of the Gallatin National Forest Avalanche Center Instructor Training | 15.11.2022 | 9 |
| Southwest Montana Ski Patrol Snow Science Day | 18.11.2022 | 30 |
| Mountain Guides Meeting, Innsbruck #1 | 30.11.2022 | 17 |
| Mountain Guides Meeting, Innsbruck #2 | 15.12.2022 | 15 |
| Forecasters at Parks Canada | 24.02.2023 | 4 |
| Colorado Avalanche Information Center | 02.03.2023 | 5 |

| | | |
|---|---|---|
| Sawtooth Avalanche Center | 08.03.2023 | 26 |
| Chugach National Forest Avalanche Information Center | 13.03.2023 | 3 |
| Gallatin National Forest Avalanche Center Professional Development Workshop | 05.04.2023 | 17 |

Appendix-2: A description of each event, date and number of samples gathered.