# Peer review of "How hard do avalanche practitioners tap during snow stability tests?"

_EGUsphere, 2023_

## Referee Comment (RC1)

**1 General remarks**

The study analyzes the (un)reliability related to the force applied during hand tap tests by almost 300 avalanche practitioners, using a device which simulates performing a snow stability test like a Compression Test (CT) or Extended Column Test (ECT). Beside the analysis of force measurements, the study also briefly addresses potential explanatory factors for observed variations in the applied force (like body height). Based on the results, the authors propose that (a) the instructions for performing these tests should be revised and that (b) the interpretation of test results should be revisited.

The subject matter of the paper primarily appeals to the snow avalanche community, particularly to avalanche practitioners. While the paper is overall generally easy to read, there is a need for reorganizing its structure. Certain sections, such as the Introduction or Methods, lack essential context or references. Additionally, parts of the discussion tend to repeat information already presented, and in some instances, they offer background that would have been beneficial if introduced earlier, i.e. in the Methods section. The technical setup, the extraction and analysis of the force measurements are clearly described, and the applied methodology seems appropriate.

The subject matter is not novel as, recently, two studies addressed practically the same topic using a (rather) comparable approach (Sedon, 2021; Griesser et al., 2023, summarized below). However, the presented study has the advantage of relying on a likewise large sample ($N = 286$ vs. $N = 69$ and $N = 62$, respectively). While the similarities in study design and results clearly decreases the novelty of the research, from my perspective, the study still warrants publication, if it were to focus more strongly on the (un)reliabilty of applying force when performing these tests, and on how to mitigate these effects. While the authors do propose potential ways forward, their propositions unfortunately remain vague rather than providing more specific, applicable recommendations. However, I feel that the authors are actually in a good position to provide data-driven actionable recommendations based on the results obtained from their own comparably large data-set, which is fully supported by the findings from the two other studies. This would considerably strengthen the submission, by adding novelty and by linking science to practice. To reach this objective, a number of changes would be necessary in the manuscript, which I explain in more detail below.

**1.1 General comments**

**1.1.1 Previous research and specification of research gap.**

My most important remark relates to the similarity of the research with the two previous studies:

– Sedon (2021) used a setup very similar to the one described here, where force was measured directly on the force measuring device. Sedon (2021) also explored the correlation between the applied force with explanatory factors like the persons' height, anger, or hand. Sedon (2021) made some propositions on how to increase consistency, as for instance by applying force dropping a ski pole from a specified height rather than using the hand to tap.

– Griesser et al. (2023), which was also co-authored by the main author, focused on measuring the stress observed in the snow, but also presented some measurements where force was applied directly to the device. The main differences between these parts of the studies seems to be the device used for measurements and the number of participants. Moreover, the data presented here was also used by Griesser et al. (2023) to compare the two devices. Moreover, Griesser et al. also explored variations in tapping force as a function of the persons' body features.

From my perspective, the methodologies (though using different devices) and findings are remarkably similar between these studies, even though absolute force measurement values differ (due to the device, I guess). For instance, the findings presented in Figure 4 are similar to the findings presented in Figure 10 in Griesser et al. (2023), and in Figure 2 in Sedon (2021). The same seems true for factors potentially explaining the variation between participants (like a persons' body height). In other words, there is considerable overlap between these three studies. Unfortunately, these studies are either not referred to (Sedon, 2021), or are only briefly mentioned (Griesser et al., 2023, L112-114). However, introducing these previous studies in greater detail is necessary to specify the research gap, and hence the research questions, and to evaluate the presented findings with regard to the mentioned previous research.

As I said above, I feel the authors could turn the similarity between studies into a strength of this submission. I provide more recommendations below.

**1.1.2 Title**

The title assumes that the reader is part of the "we"? How about something like: "Addressing the (un)reliability of force applications during snow stability tests".

**1.1.3 Introduction - Section 1**

The Introduction seems to assume that the reader is already fairly familiar with stability tests, what they are and how they work. However, most NHESS readers are likely not familiar with these tests. Therefore the Introduction requires more general background.

For instance, the introduction starts with the definition of snowpack stability (L22) rather than by introducing avalanche hazard in a more general context, leading to the factors, which define avalanche hazard (like snow stability), followed by what

snow stability is, and how it can be assessed. The latter point is addressed on L36-41. In contrast, I am not sure whether introducing the four classes of snowpack stability, the matrix used by forecasters to assess the danger level, and an example for using stability classes in the matrix (L26-34) is really relevant for the topic of increasing the reliability of hand tap tests.

Two stability tests, the Compression Test (L49) and Extended Column Test (L54) are introduced, without explaining what these tests aim to detect, and how this is being achieved. Regarding the first of these two points, Birkeland et al. (2023) provides a nice summary of the questions being addressed when performing these tests, which may be useful: «1) Is there a slab over a weak layer?, 2) Can we initiate a failure in the weak layer?, and 3) Will the crack propagate?» Moreover, a description and figure displaying the dimensions of these two tests would be helpful before listing the tapping instructions. Such a figure could also facilitate explaining the compression of snow when tapping, fracture initiation in a weak layer, and fracture propagation.

L48: please make the distinction between Rutschblock test and CT/ECT clearer (one is loaded by the weight of a human, the others are hand tap tests).

L63-83: Three examples for tapping instructions are provided. I suggest mentioning that these are examples, as instructions may vary even more, as, for instance in Switzerland, where the instructions are simply: «The blade of the avalanche shovel is placed on the block on one side and successively loaded with 10 hits each from the wrist (01-10), the elbow (11-20) and the shoulder (21-30).» (Dürr and Darms, 2016, p. 46). Consider shortening the instructions, as the American description is included to 100% in the Canadian instruction, though the latter contains an additional sentence for elbow and shoulder taps (which you could highlight using italics, if you like).

L84-102: It is certainly helpful to provide a brief summary of current approaches to interpret test results. However, I wonder whether the level of detail regarding the interpretation schemes proposed by Winkler and Schweizer (2009) and Techel et al. (2020b) is necessary for this study, or whether this could be summarized with fewer words. Again, it may be helpful to refer to the recent summary of stability tests by Birkeland et al. (2023). There are also some studies, which explored the repeatability of obtaining a similar test result at the scale of a snow pit, as for instance for the CT, which is most related to your setup (Schweizer and Bellaire, 2010).

L109-114: Please provide more detail on this previous research as it is highly relevant for your study, permitting you to address open questions and or deficiencies in these studies, and, thus, specifying the research gap and, consequently, your research questions.

L116-119: I suggest rephrasing the objectives of your study given my previous comment.

L118-119, L121-126: When reading the manuscript, it was rather surprising that you suddenly address the objective of providing data for mathematical modeling of stability tests, which was not mentioned before. If this is really an objective please be more specific why it would be necessary to model stability tests, and where the research gap is, which you are trying to address. Introduce the research gap before you bring this objective.

**1.2 Methods - Section 2**

*Section 2.1 The device: tap-o-meter* introduces the technical details regarding the developed device. I wonder whether L308-317 could be moved from the Discussion and could be integrated into the respective paragraphs in this section. I feel this applies also to L298-306, which, however, could be shortened considerably.

*Section 2.2 Data collection process* L163-166. Please provide further details: Did you provide instructions prior to tapping experiments? If so, what were these instructions? Please mention the date and number of participants at each of the events, maybe in a small table. Show the total number of participants (if I am not mistaken, you only mention this number in the abstract?). After all, the sample size is a strength of your study.

*Section 2.2 Data processing* Mention in Figure 2 that this is an example. Consider removing the second and third sentence from the caption, as this information is provided in the text (L208-209).

I feel that *Section 4.1.3 Metric Selection* could be integrated into *Section 2.3*, as it describes why peak force and loading rate are chosen and not other metrics, which are introduced in this section.

I suggest adding another subsection titled *Statistical analysis*, or similar, where the modeling approach could be described (L255). In this section you could also introduce the statistical test you are using, and what $p$-value is considered as significant.

**1.3 Results - Section 3**

**1.3.1 Tapping force**

Currently, in the text (L220-240) you repeat most of the results, which are also shown in Tables 1 and 2. As a reader, I feel this is not very informative. Instead, I propose to select and highlight the key findings like that the median force doubles from one class to the next. This doubling of force values from one class to the next is an interesting result. It is also fully in line with Sedon (2021) and Griesser et al. (2023). I would make this much more obvious to the reader.

I propose to combine L220-240 with Section 3.1, as it all relates to force measurements.

Regarding these results, which I believe represent the key findings of the study, I allow myself to propose an alternative way to present the results (Figure 1). Maybe a Figure similar to this could assist in summarizing the results, when proposing a «normal» or typical force for each tapping level, and, may thus help to highlight deviations from the «normal» , and what such deviations may mean: You have a data set comprising about 2800 taps per tapping class, from avalanche professionals from different countries (which I would emphasize as an additional strength of the data set). Given your data, you could define a «tapping norm» according to the force applied by the majority (you already mention something like this on L359), by choosing optimal thresholds between classes *wrist*, *elbow*, and *shoulder*. In your case, these may be close to the IQR shown in Table 1 and Figure 4. In addition, there could be two classes with taps lower than the desired minimal value for *wrist* taps (<*wrist*) and higher than the desired maximal value for *shoulder* taps (>*shoulder*). You could then show the proportions of observed cases in each bin. - This would be a very visual way to show the distribution of the data, the frequency distribution and magnitude of «errors» (or variation) in terms of their correspondence with a majority force «norm». Moreover, you could use such a figure to describe the proportion of participants who had all their taps lower (i.e., cells with red cross) or higher (cells with blue

[Figure]

**Figure 1.** Proposition to visualize, analyze and discuss results on tapping force. Cells could be highlighted to mark the force, which should ideally be applied.

circle) than what you would define «normal». In other words, participants tapping with a force falling into the red-cross cells, essentially lack an entire level of force values when performing the test, while participants tapping with force values in the blue-circle cells, add ten taps harder than the force normal for shoulder taps. (maybe again twice as hard?). This will obviously impact snow compaction, and, hence, the force exerted on potential weak layers.

**1.3.2 Survey**

In Section 3.2, the method of multivariate regression is mentioned (L255). Please introduce this method with sufficient detail in the Methods section.

On L361-370, some results are described, e.g. «... weight and height are significantly and positively correlated with tap force...». However, no numbers are presented. If possible, please provide more detail on the results.

As the model(s) seemed to have little explanatory power (though again no numbers are shown) (L366), a bi-variate analysis was performed. I suggest to compile the results from this analysis in a table, showing, for instance, median values for each analysed group and tap level, and whether these were significantly different. This table could either be presented in the main part of the paper, or in the Appendix. Presenting these findings in a more accessible manner will also distinguish your study from Sedon (2021) and Griesser et al. (2023), who also primarily mentioned these results briefly in the text. Moreover, it will be easier to refer to specific results when taken them up and comparing them with the respective findings presented by Sedon (2021) and Griesser et al. (2023) in the Discussion section.

**1.4 Discussion - Section 4**

**1.4.1 Discussion of limitations due to force measuring device**

In Section 4.1, the force measuring device is being discussed. Several paragraphs could probably be moved to the Methods section, as proposed before. Instead, it could be discussed that the use of different devices likely leads to differences in absolute force values in the three studies.

To decrease variability in applied force during tapping, you propose that a training device could be developed, providing participants with feedback on their tapping during training sessions (L406-408). This is a useful proposition. However, as there is no standard for building such a device, potentially leading to variations in measured force due to the device, it would be valuable if you could provide appropriate drop heights using a specific weight to obtain the median force values obtained for the three tapping classes. If new devices were being built, their sensitivity to force measurements could be compared to your device, which would allow integrating your recommendations on typical force values associated with *wrist*, *elbow*, and *shoulder* taps.

To achieve more consistent tapping in the field, you briefly mention that known weights could be used (L408-409). This is also in line with the proposition by Sedon (2021). Again, it may be worthwhile to provide specific recommendations, like the actual drop height of a ski pole, needed to achieve a desired impact force.

**1.4.2 Comparison with findings in other studies**

I suggest to include a subsection where you compare your results with the findings in Sedon (2021) and Griesser et al. (2023). For instance, you could show a small table with the median force values from the three studies. As all three studies show an approximate doubling of the force from one tapping class to the next, you could propose an addition to the instructions (L66-82) by a statement like «Tapping should be about twice as hard for elbow taps as for wrist taps.»This would be a very practical advice, and, potentially, be more useful than the Canadian description cited on L72-73.

I also suggest to briefly summarize and compare the findings from the corresponding surveys relating age, body size, ... to tapping force. Again, this may potentially lead to some recommendations for practice.

You introduce the instructions taken from the observation guidelines in detail, which had me expect that there are differences between U.S., Canadian, and Norwegian avalanche professionals. However, this didn't seem to be the case. Please take up and discuss this finding. What does this apply for the interpretation of test results?

On L15-17, L370-371, and at a few more places, you emphasize that the variability of tapping force between participants questions the reliability of these results. Please discuss this in more detail. Please also provide more detail on why your interpretation of the results is different to Griesser et al. (2023, p. 6) who obtained very similar results but concluded: «We could show that the differences between different test persons was surprisingly narrow, ...» I feel this different view on results is particularly interesting as the main author participated in both studies.

**1.4.3 Discussion of impact force**

L355-357: You mention that in Canada, there is no advice on how hard one should tap. However, neither of the other two instructions shown on L63-83 state these. Please rephrase.

L358-359: As outlined above, instead of mentioning that such an approach could be undertaken, I propose to do so, and show the results. I feel this would considerably strengthen your manuscript.

**1.4.4 Ways forward**

To me, *Section 5.1 Calls to Action*, currently in the Conclusions, should be moved to the Discussion section.

Assuming that you show more clearly the limitations due to variability in tapping force, I agree that it is important to propose result-driven ways forward. Consider whether making it more clear that the proposition of reducing tapping force variability through the use of training and an appropriate tool (Sect. 5.1.1) is the more relevant proposition compared to the second way forward. As mentioned before, I also suggest taking up the proposition by Sedon (2021), who proposed dropping a ski pole or another piece of equipment normally carried in the field from a certain height to achieve more reliable tapping.

I believe that your data, combined with the two other studies, allows you to make recommendations to the tapping instructions. Doing so would make a nice link from science to practice.

Regarding your second proposition, - reducing the interpretation of test results by excluding the tapping force (L410-419), please provide a more thorough discussion. Please discuss why this proposition may be warranted, and why not. For instance, in the two cited studies (Winkler and Schweizer, 2009; Techel et al., 2020b), which relied on Swiss data (dozens of different observers) but also in Techel et al. (2020a), which in addition made use of North American data (probably hundreds of different observers), tapping force showed to be a relevant discriminator, though (clearly) at a much lower level than fracture propagation. Moreover, reducing the interpretation of ECT test results to fracture propagation is prone to misinterpretations too: for instance, Techel et al. (2016) showed that side-by-side ECT showed contradictory propagation results 20% of the time. - For these reasons, please provide a more in-depth discussion of your proposition. And lastly, along this line, it may be worth repeating, what Birkeland et al. (2023) wrote: «Stability tests provide important data for stability evaluations during times of conditional stability. However, no test provides a definitive *go/no go* result. With accuracies of around 80%, tests are obviously not reliable enough to bet your life on them.»

**1.4.5 Limitations**

I suggest to shorten and move several parts of the Discussion to a Limitations section. This may include, for instance, L326-332.

**1.5 Abstract and Conclusions**

I suggest rephrasing Abstract and Conclusions after revising the manuscript.

**References**

Birkeland, K. W., van Herwijnen, A., Techel, F., Bair, E. H., Reuter, B., Simenhois, R., Jamieson, B., Marienthal, A., Chabot, D., and Schweizer, J.: Comparing stability tests and understanding their limitations, in: Proceedings International Snow Science Workshop, Bend, OR, USA, 2023.

Dürr, L. and Darms, G.: SLF-Beobachterhandbuch (Observation guidelines), WSL Institute for Snow and Avalanche Research SLF, Davos, https://www.slf.ch/fileadmin/user_upload/WSL/Publikationen/Sonderformate/pdf/SLF-Beobachterhandbuch.pdf, 2016.

Griesser, S., Pielmeier, C., Boutera Toft, H., and Reiweger, I.: Stress measurements in the weak layer during snow stability tests, Annals of Glaciology, p. 1–7, https://doi.org/10.1017/aog.2023.49, 2023.

Schweizer, J. and Bellaire, S.: On stability sampling strategy at the slope scale, Cold Regions Science and Technology, 64, 104–109, https://doi.org/10.1016/j.coldregions.2010.02.013, 2010.

Sedon, M.: Evaluating forces for extended column tests and compression tests, 127, 39–41, 2021.

Techel, F., Walcher, M., and Winkler, K.: Extended Column Test: repeatability and comparison to slope stability and the Rutschblock, in: Proceedings ISSW 2016. International Snow Science Workshop, 2–7 October 2016, Breckenridge, Co., pp. 1203–1208, 2016.

Techel, F., Birkeland, K., Chabot, D., Earl, J., Moner, I., and Simenhois, R.: Comparing Extended Column Test results to signs of instability in the surrounding slopes - exploring a large, international data set, The Avalanche Review, 39, 24–25, 2020a.

Techel, F., Winkler, K., Walcher, M., van Herwijnen, A., and Schweizer, J.: On snow stability interpretation of extended column test results, Natural Hazards Earth System Sciences, 20, 1941–1953, https://doi.org/10.5194/nhess-2020-50, 2020b.

Winkler, K. and Schweizer, J.: Comparison of snow stability tests: Extended Column Test, Rutschblock test and Compression Test, Cold Regions Science and Technology, 59, 217–226, https://doi.org/10.1016/j.coldregions.2009.05.003, 2009.

---

## Referee Comment (RC2)

[referee-annotated manuscript omitted]

---

## Author Comment (AC1)

**Response to reviewer comments**

Toft, H. B., Verplanck, S. V., and Landrø, M.: How hard do we tap during snow stability tests?, EGUsphere [preprint], https://doi.org/10.5194/egusphere-2023-2921, in review, 2023.

Responses are provided using red text.

**Review 1 – Frank Techel**

**1   General remarks**

The study analyzes the (un)reliability related to the force applied during hand tap tests by almost 300 avalanche practitioners, using a device which simulates performing a snow stability test like a Compression Test (CT) or Extended Column Test (ECT). Beside the analysis of force measurements, the study also briefly addresses potential explanatory factors for observed variations in the applied force (like body height). Based on the results, the authors propose that (a) the instructions for performing these tests should be revised and that (b) the interpretation of test results should be revisited.

The subject matter of the paper primarily appeals to the snow avalanche community, particularly to avalanche practitioners. While the paper is overall generally easy to read, there is a need for reorganizing its structure. Certain sections, such as the Introduction or Methods, lack essential context or references. Additionally, parts of the discussion tend to repeat information already presented, and in some instances, they offer background that would have been beneficial if introduced earlier, i.e. in the Methods section. The technical setup, the extraction and analysis of the force measurements are clearly described, and the applied methodology seems appropriate.

The subject matter is not novel as, recently, two studies addressed practically the same topic using a (rather) comparable approach (Sedon, 2021; Griesser et al., 2023, summarized below). However, the presented study has the advantage of relying on a likewise large sample (N = 286 vs. N = 69 and N = 62, respectively). While the similarities in study design and results clearly decreases the novelty of the research, from my perspective, the study still warrants publication, if it were to focus more strongly on the (un)reliabilty of applying force when performing these tests, and on how to mitigate these effects. While the authors do propose potential ways forward, their propositions unfortunately remain vague rather than providing more specific, applicable recommendations. However, I feel that the authors are actually in a good position to provide data-driven actionable recommendations based on the results obtained from their own comparably large data-set, which is fully supported by the findings from the two other studies. This would considerably strengthen the submission, by adding novelty and by linking science to practice. To reach this objective, a number of changes would be necessary in the manuscript, which I explain in more detail below.

Thank you for your thorough review. We found your comments very constructive. We have tried our best to revise our manuscript in accordance with your suggestions. In the places where we disagree, we have tried to provide a thorough explanation of why. We hope that you will find that the responses sufficient for publication. If you should find that our responses are insufficient, we welcome further feedback on how to improve it.

**1.1   General comments**

**1.1.1   Previous research and specification of research gap.**

My most important remark relates to the similarity of the research with the two previous studies:

- Sedon (2021) used a setup very similar to the one described here, where force was measured directly on the force measuring device. Sedon (2021) also explored the correlation between the applied force with explanatory factors like the persons' height, anger, or hand. Sedon (2021)

made some propositions on how to increase consistency, as for instance by applying force dropping a ski pole from a specified height rather than using the hand to tap.

- Griesser et al. (2023), which was also co-authored by the main author, focused on measuring the stress observed in the snow, but also presented some measurements where force was applied directly to the device. The main differences between these parts of the studies seems to be the device used for measurements and the number of participants. Moreover, the data presented here was also used by Griesser et al. (2023) to compare the two devices. Moreover, Griesser et al. also explored variations in tapping force as a function of the persons' body features.

From my perspective, the methodologies (though using different devices) and findings are remarkably similar between these studies, even though absolute force measurement values differ (due to the device, I guess). For instance, the findings presented in Figure 4 are similar to the findings presented in Figure 10 in Griesser et al. (2023), and in Figure 2 in Sedon (2021). The same seems true for factors potentially explaining the variation between participants (like a persons' body height). In other words, there is considerable overlap between these three studies. Unfortunately, these studies are either not referred to (Sedon, 2021), or are only briefly mentioned (Griesser et al., 2023, L112-114). However, introducing these previous studies in greater detail is necessary to specify the research gap, and hence the research questions, and to evaluate the presented findings with regard to the mentioned previous research.

We will introduce these studies more in detail in the revised manuscript. We will also include a new subsection in the discussion comparing our results with Griesser et al. (2023) and Sedon (2021). We will focus on the relative difference as you suggest, as absolute values of peak force are not comparable (see explanation for sampling rate).

As I said above, I feel the authors could turn the similarity between studies into a strength of this submission. I provide more recommendations below.

We agree that we did not present, nor compare the similarity between these studies as we should have in the initial version of the manuscript. We believe that there are some important differences, and we would like to note some of our main contributions below (more details in the specific comments).

Sample size and geographic distribution.

Griesser et al. (2023) measured 62 participants at the European Avalanche Warning Services (EAWS) conference in Davos, Switzerland. Sedon (2022) measured 69 participants from the Southern Hampshire Alpine Snow Conference in Wanaka, New Zealand. Sedon's study was not published in a peer-reviewed journal. Our study has a larger sample size of 286 participants spanning the Alps, Scandinavia, western United States, and Canada. Thus, providing a more varied sample when it comes to avalanche climate (where the test is used) and differences in avalanche specific training (how the test is performed).

Higher sampling rate (100 Hz vs. 50 kHz).

We use a higher sampling rate which enables us to measure the peak force more accurately. This is the most likely reason why we see so large differences in mean values from peak force compared to Griesser and Sedon, see a more detailed description in the specific comments.

Survey results.

Sedon and Griesser et al both test for effects of body characteristics. Their analyses consist of bivariate tests, i.e., testing if people who are heavier tap harder, and if people who are taller tap harder. A problem with this approach is that, since height and weight are correlated, the tests do not reveal which of the two factors that are more important, or if height (weight) affect tap hardness at a given weight(height). We test for effects of individual characteristics in a multivariate regression. Like Sedon and Griesser et al, we find significant effects, if we only include weight or height. However, when we add gender, these effects disappear. In other words, gender appears to be the most important factor. Due to our small sample size, statistically speaking, and the risk for low power and attenuated effects, we refrain from drawing too far-reaching conclusions from this result.

Observation of impact duration/loading rate.

An additional novelty in our study is an observation of loading rate during hand taps. It has been established that snow's response depends on the loading rate (Shapiro et al., 1997). Thus, peak force alone is not enough information to accurately model snow's response dynamic loads. Snow subject to shorter duration impacts (~1 ms) have been observed to attenuate at shallower depths than show subject to longer duration impacts (~10 ms) (Verplanck and Adams, 2023). Loading rate has also been shown to influence crack initiation in weak layers of snow (Reiweger et al. 2015). Furthermore, observing both peak force and loading rate will enable the specification of a boundary condition for future modeling efforts of the CT and ECT.

Verplanck, S. V., & Adams, E. E. (2023). *Stress Waves Through Snow Columns*. International Snow Science Workshop, Bend, Oregon.

Reiweger, I., Gaume, J., & Schweizer, J. (2015). A new mixed-mode failure criterion for weak snowpack layers. *Geophysical Research Letters*, *42*(5), 1427–1432. https://doi.org/10.1002/2014GL062780

**1.1.2    Title**

The title assumes that the reader is part of the "we"? How about something like: "Addressing the (un)reliability of force applications during snow stability tests".

We will change the title to "How hard do avalanche practitioners tap during snow stability tests?" to avoid using the term "we". The reasoning behind our title selection is that our study is focused on observation of how hard practitioners tap during snow stability tests, not addressing the reliability of snow stability tests.

**1.1.3    Introduction - Section 1**

The Introduction seems to assume that the reader is already fairly familiar with stability tests, what they are and how they work. However, most NHESS readers are likely not familiar with these tests. Therefore the Introduction requires more general background.

We agree that the introduction could benefit from a broader explanation of stability tests. We will improve the introduction section by expanding with more general background on stability tests.

For instance, the introduction starts with the definition of snowpack stability (L22) rather than by introducing avalanche hazard in a more general context, leading to the factors, which define avalanche hazard (like snow stability), followed by what 2 snow stability is, and how it can be assessed. The latter point is addressed on L36-41. In contrast, I am not sure whether introducing the four classes of snowpack stability, the matrix used by forecasters to assess the danger level, and an example for using stability classes in the matrix (L26-34) is really relevant for the topic of increasing the reliability of hand tap tests.

We did not describe the intent of our paper as clearly as intended, thank you for proving this feedback.

The intent of our paper is not to increase the reliability of hand tap tests, but rather to characterize the impact curve and quantify the variability of the hand tap loading method. This variability may affect the tests results and thus, our understanding of the stability of the snow cover. Ultimately, this can influence the forecasted avalanche danger level and backcountry decision making. We will add some sentences to reinforce this message.

Two stability tests, the Compression Test (L49) and Extended Column Test (L54) are introduced, without explaining what these tests aim to detect, and how this is being achieved. Regarding the first of these two points, Birkeland et al. (2023) provides a nice summary of the questions being addressed when performing these tests, which may be useful: «1) Is there a slab over a weak layer?, 2) Can we initiate a failure in the weak layer?, and 3) Will the crack propagate?» Moreover, a description and figure displaying the dimensions of these two tests would be helpful before listing the tapping instructions. Such

a figure could also facilitate explaining the compression of snow when tapping, fracture initiation in a weak layer, and fracture propagation.

We will add a diagram of these two tests to the introduction.

L48: please make the distinction between Rutschblock test and CT/ECT clearer (one is loaded by the weight of a human; the others are hand tap tests).

We will make this distinction between RB and CT/ECT loading in the revised manuscript.

L63-83: Three examples for tapping instructions are provided. I suggest mentioning that these are examples, as instructions may vary even more, as, for instance in Switzerland, where the instructions are simply: «The blade of the avalanche shovel is placed on the block on one side and successively loaded with 10 hits each from the wrist (01-10), the elbow (11-20) and the shoulder (21-30).» (Dürr and Darms, 2016, p. 46). Consider shortening the instructions, as the American description is included to 100% in the Canadian instruction, though the latter contains an additional sentence for elbow and shoulder taps (which you could highlight using italics, if you like).

We will mention that these are examples, include the Swiss instruction and shorten where possible.

L84-102: It is certainly helpful to provide a brief summary of current approaches to interpret test results. However, I wonder whether the level of detail regarding the interpretation schemes proposed by Winkler and Schweizer (2009) and Techel et al. (2020b) is necessary for this study, or whether this could be summarized with fewer words. Again, it may be helpful to refer to the recent summary of stability tests by Birkeland et al. (2023). There are also some studies, which explored the repeatability of obtaining a similar test result at the scale of a snow pit, as for instance for the CT, which is most related to your setup (Schweizer and Bellaire, 2010).

The ECT and CT are tests that are used extensively by practitioners in the field. Our results have implications for the conclusions drawn from test results. Since practitioners are the ones drawing these conclusions, we have tried to write the paper in a way that it is easy to read. We believe the evolution of the interpretation of the ECT test is important background information in this context. We have therefore chosen to keep the description in the text. If you disagree, we are willing to shorten it. We will refer to the recent summary from Birkeland et al. (2023) in the revised manuscript.

L109-114: Please provide more detail on this previous research as it is highly relevant for your study, permitting you to address open questions and or deficiencies in these studies, and, thus, specifying the research gap and, consequently, your research questions.

We will expand on this by including more details on the study from Logan, Griesser and the descriptive study from Sedon.

L116-119: I suggest rephrasing the objectives of your study given my previous comment.

The topic of our paper is not to increase the reliability of hand taps test, but rather to characterize the impact curve and quantify the variability of the hand tap loading method. We propose some ideas for further discussion in the "calls to action" section, which we plan to move to the discussion, but making hand tap tests more reliable is out of the scope of the study.

L118-119, L121-126: When reading the manuscript, it was rather surprising that you suddenly address the objective of providing data for mathematical modeling of stability tests, which was not mentioned before. If this is really an objective, please be more specific why it would be necessary to model stability tests, and where the research gap is, which you are trying to address. Introduce the research gap before you bring this objective.

We will address this earlier in the revised manuscript by including a paragraph to the effect of:

Stability tests are meant to simulate the avalanche release process on a small scale. The key to connecting stability tests with slope-wide avalanche mechanics is a mathematical model of the stability test. To date,

most of this has been done with the PST. A key component of the ECT is the hand-tap loading which creates a boundary condition for a mathematical model of the ECT. Although this paper does not attempt to create a mathematical model of the ECT, characterizing the impact curves are an important step towards modeling the ECT.

**1.2    Methods - Section 2**

*Section 2.1 The device: tap-o-meter* introduces the technical details regarding the developed device. I wonder whether L308- 317 could be moved from the Discussion and could be integrated into the respective paragraphs in this section. I feel this applies also to L298-306, which, however, could be shortened considerably.

We will move these sentences to the discussion as suggested.

Section 2.2 Data collection process L163-166. Please provide further details: Did you provide instructions prior to tapping experiments? If so, what were these instructions? Please mention the date and number of participants at each of the events, maybe in a small table. Show the total number of participants (if I am not mistaken, you only mention this number in the abstract?). After all, the sample size is a strength of your study.

We will expand section 2.2 by including what instructions were given, and a table with date and number of participants for each event (and total). We did not have a script. We told participants to select a glove (or no glove) to what they would use in the field, visualize themselves in a snowpit, and tap like they were doing an ECT.

Section 2.2 Data processing Mention in Figure 2 that this is an example. Consider removing the second and third sentence from the caption, as this information is provided in the text (L208-209).

We will mention that this is an example. Figure captions should be descriptive enough to stand by themselves, and in this case a few sentences were needed. We will shorten the figure texts if the reviewer has a strong opinion that it should be otherwise.

I feel that Section 4.1.3 Metric Selection could be integrated into Section 2.3, as it describes why peak force and loading rate are chosen and no other metrics, which are introduced in this section.

We will move section 4.1.3 to section 2.3 as suggested.

I suggest adding another subsection titled Statistical analysis, or similar, where the modeling approach could be described (L255). In this section you could also introduce the statistical test you are using, and what p-value is considered as significant.

We will include a short subsection on the statistical methods in the revised manuscript (simple OLS, t-tests and 5% p-value threshold).

**1.3    Results - Section 3**

**1.3.1    Tapping force**

Currently, in the text (L220-240) you repeat most of the results, which are also shown in Tables 1 and 2. As a reader, I feel this is not very informative. Instead, I propose to select and highlight the key findings like that the median force doubles from one class to the next. This doubling of force values from one class to the next is an interesting result. It is also fully in line with Sedon (2021) and Griesser et al. (2023). I would make this much more obvious to the reader.

I propose to combine L220-240 with Section 3.1, as it all relates to force measurements.

We will rewrite L220-240 and focus on key findings as suggested and integrate it with section 3.1.

Regarding these results, which I believe represent the key findings of the study, I allow myself to propose an alternative way to present the results (Figure 1). Maybe a Figure similar to this could assist in summarizing the results, when proposing a «normal» or typical force for each tapping level, and, may thus help to highlight deviations from the «normal», and what such deviations may mean: You have a data set comprising about 2800 taps per tapping class, from avalanche professionals from different countries (which I would emphasize as an additional strength of the data set). Given your data, you could define a «tapping norm» according to the force applied by the majority (you already mention something like this on L359), by choosing optimal thresholds between classes wrist, elbow, and shoulder. In your case, these may be close to the IQR shown in Table 1 and Figure 4. In addition, there could be two classes with taps lower than the desired minimal value for wrist taps (shoulder). You could then show the proportions of observed cases in each bin. - This would be a very visual way to show the distribution of the data, the frequency distribution and magnitude of «errors» (or variation) in terms of their correspondence with a majority force «norm». Moreover, you could use such a figure to describe the proportion of participants who had all their taps lower (i.e., cells with red cross) or higher (cells with blue circle) than what you would define «normal». In other words, participants tapping with a force falling into the red-cross cells, essentially lack an entire level of force values when performing the test, while participants tapping with force values in the blue-circle cells, add ten taps harder than the force normal for shoulder taps. (maybe again twice as hard?). This will obviously impact snow compaction, and, hence, the force exerted on potential weak layers.

[Figure]

Figure 1. Proposition to visualize, analyze and discuss results on tapping force. Cells could be highlighted to mark the force, which should ideally be applied.

Thank you for an interesting idea. While both visualizations have their place, we believe the boxplots are generally more intuitive for a broad audience due to their simplicity and familiarity. While confusion matrices have their own strengths in showing relationships between the "majority force norm", they might not be as easy to interpret as practitioners are less familiar with them. However, we do think the table you suggested could help us demonstrate the overlap between tap levels even though the tap levels are statistically different and will therefore include it in addition to figure 4.

Table example:

|  | < 51 N | 51 – 125 N | 125 – 245 N | 245 – 505 N | > 505 N |
|---|---|---|---|---|---|
| Wrist | 24.74% | 58.09% | 16.42% | 0.74% | 0.00% |

| | | | | | |
|---|---|---|---|---|---|
| Elbow | 1.02% | 23.99% | 49.77% | 23.85% | 1.37% |
| Shoulder | 0.03% | 3.10% | 22.07% | 49.75% | 25.05% |

**1.3.2 Survey**

In Section 3.2, the method of multivariate regression is mentioned (L255). Please introduce this method with sufficient detail in the Methods section.

We will introduce the method in the methods section in the revised manuscript.

On L361-370, some results are described, e.g. «... weight and height are significantly and positively correlated with tap force...». However, no numbers are presented. If possible, please provide more detail on the results.

We will expand on this in the revised manuscript with p-values in text and tables in the appendix. The reason we did not include these details in the original manuscript was that we did not want to emphasize these results as they were not significant when using a multivariate model. There is correlation, but when including gender in the multivariate model, the weight and height lost their significance.

As the model(s) seemed to have little explanatory power (though again no numbers are shown) (L366), a bi-variate analysis was performed. I suggest to compile the results from this analysis in a table, showing, for instance, median values for each analysed group and tap level, and whether these were significantly different. This table could either be presented in the main part of the paper, or in the Appendix. Presenting these findings in a more accessible manner will also distinguish your study from Sedon (2021) and Griesser et al. (2023), who also primarily mentioned these results briefly in the text. Moreover, it will be easier to refer to specific results when taken them up and comparing them with the respective findings presented by Sedon (2021) and Griesser et al. (2023) in the Discussion section.

We will add supplementary tables for the different statistical analysis performed in the appendix and include statistical numbers in the survey section.

**1.4 Discussion - Section 4**

**1.4.1 Discussion of limitations due to force measuring device**

In Section 4.1, the force measuring device is being discussed. Several paragraphs could probably be moved to the Methods section, as proposed before. Instead, it could be discussed that the use of different devices likely leads to differences in absolute force values in the three studies.

We will move some paragraphs to the methods section in the revised manuscript as suggested earlier.

We will include a paragraph about why the difference in sampling rate likely leads to differences in absolute peak force values in the three studies.

To explain the concept, we have allowed ourselves to provide a visual example below for the reviewer response. We do not plan to include this figure in the revised manuscript.

We don't believe the differences in peak force measurements is due to the device itself, but rather the sampling rate. To calculate what an adequate sampling rate is, we need to know the loading duration of the signal we would like to measure.

We provide a simplified visual example using a gaussian curve with a loading duration of ~0.025 s. We compare sampling rates between 125 Hz and 1,000 Hz for a hypothetic impact curve. In this example, a sampling rate of 125 Hz would measure the peak force to be between 20-125 N (~70 N on average),

while a sampling rate of 1,000 Hz would measure 125 N. This example highlights the importance of sampling rate if we are interested in accurate impact force measurements that can be used to compare different individuals, or between devices.

[Figure]

The figure shows four examples of sampling rate from 125 to 1,000 Hz. Lower sampling rates are less likely to accurately capture the peak value as its random where the measurement is being made on the curve. The average value for each sample rate is noted in red (bold).

Because Griesser et al. 2023 are not able to accurately measure the peak forces (unknown for Sedon, 2020), they are also not able to accurately measure loading rate with their devices' lower sampling rate (which differs for different tap levels).

We have written this about our reasoning for our sampling rate in L157-161:

*"To determine an appropriate sampling rate, knowledge of the impact signal is critical. We are most interested in the peak force and loading rate leading up to it. Preliminary testing showed that this rise time is fastest for the shoulder taps and can happen as quickly as a few milliseconds. Conservatively assuming this rise occurs over 1 millisecond, a sampling rate of 50 kHz leads to 50 samples in this critical measurement period. A number deemed sufficient for our purposes and within the capabilities of the measurement system."*

To decrease variability in applied force during tapping, you propose that a training device could be developed, providing participants with feedback on their tapping during training sessions (L406-408). This is a useful proposition. However, as there is no standard for building such a device, potentially leading to variations in measured force due to the device, it would be valuable if you could provide appropriate drop heights using a specific weight to obtain the median force values obtained for the three tapping classes. If new devices were being built, their sensitivity to force measurements could be compared to your device, which would allow integrating your recommendations on typical force values associated with wrist, elbow, and shoulder taps.

If a training device is considered to be the best solution to reduce tapping variability, we hope that our methods section includes sufficient information to build such a training device.

The scope of our paper is to quantify how hard practitioners tap, not how hard they should tap. We will make this clearer in the revised manuscript.

Lastly, it would be challenging to suggest drop heights for specific objects. The peak force and loading rate are coupled and depend on the object's mass, the drop height, and the materials that are in contact during impact. Not only mass and height would need to be recommended, but also materials and possible use of cushion like material to recreate both peak force and loading rate of hand taps. Verplanck and Adams (2023) attempted to match the impact curves of hand taps using an acetal mass, foam cushion, and aluminum plate. However, they attempted to match their own hand taps, not the averages presented in our study.

To achieve more consistent tapping in the field, you briefly mention that known weights could be used (L408-409). This is also in line with the proposition by Sedon (2021). Again, it may be worthwhile to provide specific recommendations, like the actual drop height of a ski pole, needed to achieve a desired impact force.

Thank you for pointing this out. A key component we did not mention here is the impact duration/loading rate. See also the explanation above. We will improve this sentence by saying that weights at specific heights could be used, but it would be challenging to standardize. We would have to define material and possibly a cushion like material to make this a viable option, matching both the peak force and loading rate of hand taps.

**1.4.2    Comparison with findings in other studies**

I suggest including a subsection where you compare your results with the findings in Sedon (2021) and Griesser et al. (2023). For instance, you could show a small table with the median force values from the three studies. As all three studies show an approximate doubling of the force from one tapping class to the next, you could propose an addition to the instructions (L66-82) by a statement like «Tapping should be about twice as hard for elbow taps as for wrist taps». This would be a very practical advice, and, potentially, be more useful than the Canadian description cited on L72-73.

We will include a new subsection comparing our results with Griesser et al. (2023) and Sedon (2021) as suggested. We will focus on the relative difference as you suggest, as absolute values of peak force are not comparable (see explanation for sampling rate). We will not include a statement on how hard taps should be as this is outside the scope of this paper.

I also suggest to briefly summarize and compare the findings from the corresponding surveys relating age, body size, ... to tapping force. Again, this may potentially lead to some recommendations for practice.

We will add a paragraph comparing the main results from the survey data and the details of the statistical analysis in the appendix. The main finding is that only gender is statistically significant. Body features is also correlated with tap force, but when included in a multivariate analysis with gender, they disappear. We believe the correlation for body features is likely due to men being in general taller, and heavier.

You introduce the instructions taken from the observation guidelines in detail, which had me expect that there are differences between U.S., Canadian, and Norwegian avalanche professionals. However, this didn't seem to be the case. Please take up and discuss this finding. What does this apply for the interpretation of test results?

We will include a section on this in the revised manuscript. We do not find any regional differences in tapping force from Scandinavia to the Alps to North America. However, our predictive power is low. This could be due to our small sample in a statistical context (n=286).

On L15-17, L370-371, and at a few more places, you emphasize that the variability of tapping force between participants questions the reliability of these results. Please discuss this in more detail. Please also provide more detail on why your interpretation of the results is different to Griesser et al. (2023, p. 6) who obtained very similar results but concluded: «We could show that the differences between different test persons was surprisingly narrow, ...» I feel this different view on results is particularly interesting as the main author participated in both studies.

To address the differences in interpretation between our findings and those of Griesser et al. (2023), we should review the data and its implications in the context of avalanche decision making where the consequence is high, and the margin of error is small.

Even though a one-way ANOVA results in a P-value lower than 0.01, indicating that there are (without doubt) statistically significant differences between the three loading steps, mirroring the findings of Griesser et al. (2023). Our interpretation is different due to the emphasis we place on the variability of tapping force across participants. This variability, even though they are statistically different, raises concerns about the reliability and applicability of the average results to individual cases, especially in scenarios with the potential for fatal outcomes.

The main difference in our argument lies in the inherent risk of relying solely on mean statistics in avalanche terrain which is a risky environment. While Griesser et al. (2023) highlights the narrow differences as a positive outcome, we focus on the implications of the differences more critically. The presence of significant overlap between the 25th-75th percentile ranges of force applied during elbow taps with those of wrist and shoulder taps, where ~24% of the data for elbow taps overlaps with wrist and shoulder taps respectively. These overlaps could have practical significance in real-world applications.

Our interpretation aligns with the principle of 'err on the side of caution,' especially in fields where the consequences of errors can be catastrophic.

We will include more details in the discussion in the revised manuscript.

**1.4.3 Discussion of impact force**

L355-357: You mention that in Canada, there is no advice on how hard one should tap. However, neither of the other two instructions shown on L63-83 state these. Please rephrase.

We will rephrase this sentence in the revised manuscript.

L358-359: As outlined above, instead of mentioning that such an approach could be undertaken, I propose to do so, and show the results. I feel this would considerably strengthen your manuscript.

The scope of our paper is to measure the magnitude and variability of impact forces from hand taps, not how hard they should be. We do provide some ideas moving forward (calls to action), where one includes standardization. One method to do this could be by defining some thresholds for hand taps, combined with training.

If we used our data to define these thresholds, we would imply that a "Wisdom of the crowds" (WoC) approach is the best way to define the correct tap levels. The risk of using WoC to define the thresholds for impact force is that they could have drifted away, or never been equal to the impact forces the authors of the test originally intended to capture. Furthermore, the "wisdom of the crowds" approach may lead to recommended tap forces which are too light, particularly for the taps from shoulder. Many avalanche practitioners are of the mind that the goal of "stability tests" is to hunt for instability. If we recommend shoulder taps that are too light, this could lead to the dangerous outcome of an increase in false stable results. We believe it should be a discussion among the tests' creators, the scientific community, and the practitioner community to define thresholds, guidelines, and test interpretation.

**1.4.4 Ways forward**

To me, Section 5.1 Calls to Action, currently in the Conclusions, should be moved to the Discussion section.

We will move section 5.1 (Calls to action) to a section in the discussion titled "Future topics of discussion for improved standards" to make it clearer to the reader that recommending how the standards should be updated specifically is out of the scope of work.

Assuming that you show more clearly the limitations due to variability in tapping force, I agree that it is important to propose result-driven ways forward. Consider whether making it more clear that the proposition of reducing tapping force variability through the use of training and an appropriate tool (Sect. 5.1.1) is the more relevant proposition compared to the second way forward. As mentioned before, I also suggest taking up the proposition by Sedon (2021), who proposed dropping a ski pole or another piece of equipment normally carried in the field from a certain height to achieve more reliable tapping.

There are challenges with both directions. It would be challenging to develop a test routine that uses equipment normally carried in the field due to the different loading rates for different tap levels. The peak force and loading rate are coupled and depend on the object's mass, the drop height, and the materials that are in contact during impact. Not only mass and height would need to be recommended, but also materials and possible use of cushion like material to recreate both peak force and loading rate of hand taps. We will elaborate on this in the revised manuscript.

I believe that your data, combined with the two other studies, allows you to make recommendations to the tapping instructions. Doing so would make a nice link from science to practice.

See response above to L358-359 (section 1.4.3, Discussion of impact force).

Regarding your second proposition, - reducing the interpretation of test results by excluding the tapping force (L410-419), please provide a more thorough discussion. Please discuss why this proposition may be warranted, and why not. For instance, in the two cited studies (Winkler and Schweizer, 2009; Techel et al., 2020b), which relied on Swiss data (dozens of different observers) but also in Techel et al. (2020a), which in addition made use of North American data (probably hundreds of different observers), tapping force showed to be a relevant discriminator, though (clearly) at a much lower level than fracture propagation. Moreover, reducing the interpretation of ECT test results to fracture propagation is prone to misinterpretations too: for instance, Techel et al. (2016) showed that side-by-side ECT showed contradictory propagation results 20% of the time. - For these reasons, please provide a more in-depth discussion of your proposition. And lastly, along this line, it may be worth repeating, what Birkeland et al. (2023) wrote: «Stability tests provide important data for stability evaluations during times of conditional stability. However, no test provides a definitive go/no go result. With accuracies of around 80%, tests are obviously not reliable enough to bet your life on them».

We will include a larger discussion around our second proposition of limiting the scope of the test.

Our second proposition comes from the implication of defining predictor thresholds based on impact forces from a large database of ECTs. The concern is that the wide range of variability in impact force observed in our study makes these average-based thresholds relatively weak indicators. The thresholds make sense when analyzing large amounts of data (e.g. in the context of avalanche forecasting), but not when applying the average results to individual cases.

One example could be whether it's valid to interpret ECTP20 differently compared to ECTP24 in individual cases, given the large discrepancies in tapping force.

We acknowledge that the interpretation of ECTP vs. ECTN is not straight forward either, but our opinion is that it is less affected by the variability of hand taps and is therefore a more consistent limitation that is unrelated to the impact force of the hand taps themselves.

Another example showing that the interpretation of ECT has gone towards a simpler approach, rather than collecting as much data as possible was the removal of shear quality and fracture characteristics from the ECT. This shows that the authors of the ECT previously have made changes to make it easier to interpret the test (Simenhois et al. 2018) at the expense of leaving potentially relevant information out.

Simenhois, R., Chabot, D., Birkeland, K. and Greene, E. (2018). Shear Quality or Fracture Character with an Extended Column Test – No Longer in SWAG or SnowPilot.

**1.4.5    Limitations**

I suggest to shorten and move several parts of the Discussion to a Limitations section. This may include, for instance, L326-332.

We will review the discussion section and move relevant parts to the limitations section as suggested.

**1.5    Abstract and Conclusions**

I suggest rephrasing Abstract and Conclusions after revising the manuscript.

We will rephrase the abstract and conclusion in the revised manuscript.

**References**

Birkeland, K. W., van Herwijnen, A., Techel, F., Bair, E. H., Reuter, B., Simenhois, R., Jamieson, B., Marienthal, A., Chabot, D., and Schweizer, J.: Comparing stability tests and understanding their limitations, in: Proceedings International Snow Science Workshop, Bend, OR, USA, 2023.

Dürr, L. and Darms, G.: SLF-Beobachterhandbuch (Observation guidelines), WSL Institute for Snow and Avalanche Research SLF, Davos, https://www.slf.ch/fileadmin/user_upload/WSL/Publikationen/Sonderformate/pdf/SLF-Beobachterhandbuch.pdf, 2016.

Griesser, S., Pielmeier, C., Boutera Toft, H., and Reiweger, I.: Stress measurements in the weak layer during snow stability tests, Annals of Glaciology, p. 1–7, https://doi.org/10.1017/aog.2023.49, 2023.

Schweizer, J. and Bellaire, S.: On stability sampling strategy at the slope scale, Cold Regions Science and Technology, 64, 104–109, https://doi.org/10.1016/j.coldregions.2010.02.013, 2010.

Sedon, M.: Evaluating forces for extended column tests and compression tests, 127, 39–41, 2021.

Techel, F., Walcher, M., and Winkler, K.: Extended Column Test: repeatability and comparison to slope stability and the Rutschblock, in: Proceedings ISSW 2016. International Snow Science Workshop, 2–7 October 2016, Breckenridge, Co., pp. 1203–1208, 2016.

Techel, F., Birkeland, K., Chabot, D., Earl, J., Moner, I., and Simenhois, R.: Comparing Extended Column Test results to signs of instability in the surrounding slopes - exploring a large, international data set, The Avalanche Review, 39, 24–25, 2020a. 210 Techel, F.,

Winkler, K., Walcher, M., van Herwijnen, A., and Schweizer, J.: On snow stability interpretation of extended column test results, Natural Hazards Earth System Sciences, 20, 1941–1953, https://doi.org/10.5194/nhess-2020-50, 2020b. Winkler, K. and Schweizer, J.: Comparison of snow stability tests: Extended Column Test, Rutschblock test and Compression Test, Cold Regions Science and Technology, 59, 217–226, https://doi.org/10.1016/j.coldregions.2009.05.003, 2009

**Review 2 – Ron Simenhois**

**2     General remarks**

Dear Håvard and co-authors, dear Editor Yves Bühler,

I thoroughly enjoyed reviewing this manuscript, which presents an impressive dataset with substantial depth. The insights derived from the data have the potential to significantly enhance our understanding of "surface-loading" stability test results.

While the manuscript holds promise, there are opportunities to enhance its readability and overall quality through a few modifications. Here are some general comments, with more specific feedback available in the attached document.

Thank you for your thorough review. We found your comments very constructive. We have tried our best to revise our manuscript in accordance with your suggestions. In the places where we disagree, we have tried to provide a thorough explanation of why. We hope that you will find that the responses sufficient for publication. If you should find that our responses are insufficient, we welcome further feedback on how to improve it.

**2.1     General comments**

**2.1.1     Introduction:**

The authors refer to several other papers. A clearer explanation of how these references relate to the current manuscript would enhance reader comprehension. A brief sentence or two elucidating the relevance of these references will improve the readability of this manuscript.

We will add some sentences to highlight the importance of each reference.

The manuscript could benefit from explicitly stating what new contributions it brings to the existing body of knowledge, building upon already published material.

This was also brought to our attention from reviewer 1. We will make it clearer what our contributions are in the revised manuscript. Please see our response to Reviewer 1 in section 1.1.1 above.

To summarize, we have a larger dataset spanning from Scandinavia, the Alps, Canada and United States. We have a sampling rate which enables an accurate measurement of peak force, as well as loading rate. Although what happens below the snow surface is out of scope of this paper, loading rate is a critical component for stress wave transmission through snow (Verplanck and Adams, 2023) and fracture initiation in a weak layer (Reiweger et al. 2015). We also provide more robust statistics by using multivariate models instead of simple regression.

Verplanck, S. V., & Adams, E. E. (2023). *Stress Waves Through Snow Columns*. International Snow Science Workshop, Bend, Oregon.

Reiweger, I., Gaume, J., & Schweizer, J. (2015). A new mixed-mode failure criterion for weak snowpack layers. *Geophysical Research Letters*, *42*(5), 1427–1432. https://doi.org/10.1002/2014GL062780

**2.1.2     Methods:**

The methods section is well-crafted and easily understandable. However, a brief explanation of why peak force and loading rate were chosen as metrics for data analysis is missing. Incorporating elements from section 4.1.3 into the methods would provide valuable context.

We will incorporate elements from section 4.1.3 and the comment above for a brief explanation of why peak force and loading rate are chosen.

**2.1.3 Results:**

Consider including simple statistical analyses to assess the significance of differences between tapping levels. This addition would strengthen the overall robustness of the results.

While the authors note a disparity in tapping force between women and men, the manuscript misses an opportunity to delve into the variability of tapping force within each gender group. Addressing this aspect would add depth to the analysis.

We will include a one-way ANOVA test to assess the significance of differences between tapping levels. (p-value was found to be <0.01). We also will include more detailed statistical results in an appendix section.

**2.1.4 Discussion:**

A concise discussion regarding the relevance of the results within the snowpack load and stability assessment would be beneficial. This addition would help readers better grasp the broader implications of the findings.

We agree that a concise discussion of the relevance of our results to snowpack load and stability assessment would benefit the reader. Although snow's response to hand-tap loading is out of the scope of the paper, we could make some brief commentary.

Regarding snowpack load, we could state that the dynamic load generated from a hand tap is superimposed on the static weight of the snowpack. The attenuation of the dynamic load as it transmits through snow is out of scope of this work, but we refer the reader to the recent work by Verplanck and Adams (2023).

Regarding stability assessment, we plan to move some of the contents section "5.1.2 Limit the test's interpretation" to a new section in the discussion where we incorporate the other reviewer's feedback on this section.

**2.2 Specific comments**

L23: How? A sentence or two about how Reuter & Schweizer describe the propensity of a snow-covered slope to avalanche will add to the readability of this section.

We will add a sentence or two about this in the revised manuscript.

L54: Consider adding Sigrist, C., and Schweizer, J., 2007. Critical energy release rates of weak snowpack layers determined in field experiments. Geophysical Research Letters, 34. L03502, doi:10.1029/2006GL028576.

We will add the citation as suggested.

L89: ECTX means inconclusive results. Consider noting this here.

We will add this to the revised manuscript.

L112: What did they find? How does tapping load affect the stress within the snowpack? What are the implications for your research?

We will expand on the studies from Griesser and Sedon as per previous comment from reviewer 1.

L114: Add a sentence or two about why, after all the studies you mention, your work still contributes to the field. I believe that the size and the variability of the data.

We will add a few sentences on why our work contributes to the field (number of samples, larger regional coverage (Scandinavia, North America, and Alps), higher sampling rate, observation of loading rate, multivariate statistical models).

L143: Consider adding a "linear" here. This can clarify some of the expatiation below.

We will add linear as suggested.

L173: Did you test this theory? did users grab the shovel handles and cause to inconsistent measurements due to torquing the cell? If yest, mention it.

Early tests during the development showed that even gentle touches on a shovel handle are picked up with our sensitive load cell. No samples in this dataset are collected with the shovel handle on. We will add a sentence on this in the revised manuscript.

L179: Can you look at the variability within a specific operation or where folks regularly dig snowpits together? Depending on the result of this question, this may be something you can mention as a suggestion for improvement in the discussion.

The sample size within a specific operation or smaller area would be too small to adjust for height, weight and gender. Even our sample size of almost 300 participants from Scandinavia, the Alps and North America, was too small to find any statistically significant differences.

L181: Why did you choose these metrics?

We will briefly summarize section 4.1.3 here and include a reference to Reiweger et al. (2015) which found that weak layer's failure criteria depended on loading rate in addition to internal stress state.

L182: This is redundant and does not add clarity to the description below.

We believe it makes sense to state our two main metrics that are pulled from our tap data (maximum force and loading rate).

L186: Add a short description or reference to the algorithm. For example, by comparing neighboring values, if you used scipy.signal.find_peaks

We will do as suggested, and yes we did use scipy.signal.find_peaks.

L204: Start of the tap or when the force exceeds 15 N?

Good point, we will specify "when the force exceeds 15 N" in the revised manuscript.

L205: Consider this equation to a more standard notation (df/dt). Also, you can general the expression to allow adaptation of your methods to other tap o-meter units and replace "15 with something like "max noise value." Then you can state that, in this case, it is 15.

Good suggestions. However, using $dF/dt$ may confuse the reader between the loading rate metric and the analytic derivative of a tap idealized as a gaussian. Thus, we plan to define loading rate as $r$. We plan to change that section to the following:

The loading rate, $r$, is defined as a linear interpolation, Eq. (2), between the maximum force, $F_{peak}$, and a threshold value greater than typical noise, $\lambda$. In our measurements, a $\lambda$ of 15 N was deemed appropriate. The difference in force is divided by the rise time, $\Delta t$, to determine the loading rate. The rise time is the difference in time between the peak force and when the threshold is crossed.

$$r = \frac{(F_{peak} - \lambda)}{\Delta t} \tag{2}$$

L205: Is there a correlation between loading rate and peak force? What is the R2 between these values? Clearly, there is no need to address this if there is nothing there, but it could add to this manuscript if there is something there.

Loading rate is positively correlated with peak force (R-squared of 0.70). We will add a sentence on this in the revised manuscript.

L223: Adding percentage her in table 1 would give a better picture of the volume of outliers.

We will add percentages to the table in the revised manuscript.

L234: The wrist average loading rate is smaller than the Std. Is this correct? If so, are there negative loading rates in the dataset? Or are there other explanations? Please check this or add an explanation of how that could be. Also, correct Table 2 if needed.

Thank you for noting this, there was an error here. We will fix this in the revised manuscript by correcting Table 2.

L250 (Figure 4): There is a significant overlap between elbow and wrist and elbow and shoulder. You need to add an ANOVA or Mann-Whitney test to quantify if these steps overlap and are statistically significant. Looking at Figure 2, this may be the case for single-user cases within the dataset. How common is it for a single person to overlap a tapping force between the different steps?

Doing a one-way ANOVA, we get: F-statistic: 4012.34 and P-value: <0.01. There is a significant difference between the groups. For how common it is for a single person to overlap between different steps, see response to reviewer 1, section 1.3.1.

L264: Wow, this is a great insight!

Thank you!

L265: Is the tap force variability more or less for women than for men?

Women have a slightly lower standard deviation but considering that they on average also tap less hard, we don't think there is any relative difference if we look at the standard deviation as a function of the mean. We will add a sentence that we investigated this but did not find any differences in variability between the gender groups.

| | Mean | | | Standard deviation | | |
|---|---|---|---|---|---|---|
| Gender | Wrist | Elbow | Shoulder | Wrist | Elbow | Shoulder |
| Male | 88 N | 205 N | 420 N | 48 N | 107 N | 214 N |
| Female | 71 N | 160 N | 315 N | 41 N | 85 N | 171 N |

L270: Mentioning more details, like the number of females and males in the study and their average height and weight, are there similar variation in tapping forces between the gender groups? will improve the overall picture you are trying to paint.

We will include some more information of survey data in the revised manuscript.

L275: What is t? is it the time from the start of the tap or from when the force exceeds 15 N? If it is from the start of the tap, why it is different from EQ 3?

Good point, it is confusing as written. $t$ is simply the time variable that the force, $F$, is a function of. We also plan to replace $s$ with $\sigma$ so it will not be confused with the unit: seconds. We plan to rewrite that section as follows:

Both the peak force, $F_{peak}$, and loading rate, $r$, are used to idealize the impact curves. First, consider the equation describing a Gaussian function of force, $F$, as a function of time, $t$.

$$F(t) = F_{peak}e^{-\frac{1}{2}\left(\frac{t-t_{peak}}{\sigma}\right)^2},$$
(3)

Where $F_{peak}$ is the peak force and $t_{peak}$ is the time at which the peak force occurs. The duration of the force curve is governed by $\sigma$, the standard deviation if the Gaussian function were to be describing a normal distribution. Since 99.7% of the curve's magnitude occurs during $6\sigma$, the duration of impact is defined $6\sigma$ in our study. Thus, the rise to peak force occurs over approximately $3\sigma$.

$$r \approx \frac{F_{peak}}{3\sigma},$$
(4)

This is an approximation rather than equality because it assumes a linear rise, rather than the non-linear Gaussian shape. However, since loading rate and peak force are the two metrics ascertained from the measured data, this approximation provides a convenient way to idealize the measured force curves. Rearranging the approximation yields

$$\sigma \approx \frac{F_{peak}}{3r},$$
(5)

And substituting this relationship for $s$ in Eq. (3) yields the Gaussian approximation used to idealize the measured force-time curves.

$$F(t) \approx F_{peak}e^{-\frac{1}{2}\left(\frac{3r(t-t_{peak})}{F_{peak}}\right)^2}$$
(6)

L301: In some cases you add Kg values and in some cases you are not. I'd recommend that you stick with N only, but if you keep the reference to Kg, do that everywhere.

We will remove kg. from the manuscript as suggested.

L364: Does this variability continue when you look at male and female groups separately? You mentioned that gender is the only explanatory variable that is significantly correlated with tap force across all multivariate 265 regression models. It makes sense to address this question.

See response for L265 above.

L372: The manuscript does not clarify the benefit of the Idealization of taps as Gaussian functions. This would be a good place to elaborate on that or remove it altogether.

You are right. As written the manuscript does not clearly state the benefits of idealizing the taps as Gaussian functions. The benefits are as follows. Figure 4 only shows the peak force. By idealizing the taps as Gaussian functions, the reader is provided a visual comparison of both loading rate and peak force. It also provides a convenient way to compare the momentum associate with each tap level. The momentum is the area under the force-time curve. Momentum calculations by direct numeric integration of the load-cell data would not be accurate because of the long, oscillatory tails which are attributed to the measurement method. Lastly, Gaussian functions are often used in wave propagation problems because they represent a continuous, smooth pulse of disturbance. By providing a method of converting the peak force and loading rate to a Gaussian function, the reader is given a steppingstone towards mathematically modeling a test like the CT or ECT.

---

## Referee Report (RR1)

Dear authors,

Please find attached some comments regarding your revised manuscript.

I have three main points of critique:

- The presentation of results should be improved, to make it easier for the reader to grasp the findings of the study. Sometimes results are exclusively shown in the appendix with little or no information about these findings in the main part of the manuscript (e.g., L294-295), sometimes findings are presented for a first time in the Discussion (e.g., L355-359).
- It should be ascertained that readers who are not fully familiar with all the previous work (e.g., L419) or how the statistical analysis was done or can be interpreted (e.g., Appendix-4 to 7) can follow the line of argumentation. Please accommodate more of an outside view when revising by providing more of an explanation.
- It is clearly important to discuss the implication of the variability in tapping force regarding the stability interpretation of tests relying on hand tapping. However, this section needs much more of a balanced discussion combining findings (L364-365) with previous research on this topic.

In the following, please find some more detailed comments including on the three points mentioned before.

I hope this feedback helps to improve the manuscript.

**Comments:**

L12-13: Consider removing "Peak forces and loading rates are the metrics chosen to quantitatively compare the data." This statement seems redundant as the two metrics are used in the following sentence, and as no additional explanation is given in this sentence.

L13: Consider mentioning that peak force approximately doubles from one loading step to the next. This seems to be a key finding.

L14: "Significant" overlap – Does this refer to statistical significance in the overlapping proportions? From L357 I understood that the distributions are significantly different, but there is no mention about the overlapping proportions being statistically different? Consider reserving "significant" only when referring to statistical significance, and reword using something like "considerable", or similar. It may also pay to mention the approximate proportions of overlap in brackets.

L16: introduce ECT abbreviation, when first used.

L22: Correct, but a bit strange that the concept of snowpack stability is introduced in the sense that it has been modelled in this way. What about: "Snowpack instability describes the propensity for a slope to avalanche (CITE). Failure initiation and crack propagation are key components of the avalanche release process (CITE)."

L24-26: It is certainly correct that you state this but it interrupts reading the introduction. Consider rephrasing or moving to a different section.

L30: Is this really the reason why stability tests were invented. Please cite the respective work where this is written. Otherwise, please rephrase more along the line that snow stability tests can support decision making in case of conditional stability (e.g., Birkeland et al, 2023).

L34: Consider to start the sentence with "In contrast, in situations…"

L35: Consider explaining that this division is based on informational entropy, by emphasizing that the mentioned signs (L34) allow direct interpretation of instability (class 1), while stability tests provide more indirect information (class 2).

L69: "simulate portions of" – maybe rephrase to "reflect the"

L72: "component of the ECT" – consider adding the CT: "component of CT and ECT"

L127: Please rephrase to accommodate that Griesser et al. (2023) not only performed tests in an indoord setting but also in the field.

L134: "improved measurement device" – please specify relative to what this improvement is intended to be.

L199: Introduce EAWS when first using this abbreviation.

L212-213: Consider removing, as it is stated in the sentence before.

L258-259: Consider removing "After this second… ".

Sect. 2.4: Consider reordering this paragraph in the way you present the results: (1) methods used to compare tapping force, (2) methods used to analyze explanatory factors. Please explain in a little more detail what ANOVA actually compares and provide a reference. (for instance, I had to look it up as I rarely use this technique). Consider bringing the second sentence first, then the first sentence. Provide references to all methods and tests.

L271: Is "trend" the right word for your analysis? Consider using an alternative word.

L272-277: this is essentially a repetition of Table 2. Consider shortening, highlighting one or key facts, and referring the reader to the table for more information.

L276: Consider removing "~0.0%" as you state that it happened once (in about 8000 tests).

L281: remove "(outliers removed using 1.5 times IQR)" as this is already introduced on L272-274

L283: "(wrist, elbow, shoulder)" – consider deleting, as the loading steps are explained several times before

L285: "each load step (i.e. loading step)" – consider removing one of these

Table 3: Explain abbreviation "Std." when first being used in the manuscript.

L289-292: This is essentially the same as the caption of Figure 4. Consider removing from text or from caption.

L294-295: Why make this statement or add this table shown in Appendix-3 if you don't explain what is shown? Moreover, I would move this way of looking at the data in a more prominent way to the Results section, as it provides another way at interpreting the data, with results which allow a simple way of summarizing the proportions of agreements and disagreements. - I suggest incorporating this into the main part of the manuscript, but at least to rephrase the current statement giving the reader at least the essence of what is shown in Appendix-3. For instance, findings like "defining non-overlapping band-widths of tapping force based on the distribution of tapping forces shown in Table Table 3 and Figure 4 showed that between 49% and 54% of the taps of a loading step were within this range. However, a share of about 1% of the taps applied a force corresponding to the range of two loading steps higher or lower". or similar, could be mentioned.

L305: "cannot explain the bulk of the variance" – Please be more specific with regard to how much these factors could explain or consider rephrasing.

L303: Consider renaming the section title to something like "Explanatory factors impacting peak force" or similar.

L303-309: For the reader it is hard to follow the line of argumentation as the reader is being referred four times to different tables in the Appendix. I suggest summarizing these four tables in a compact table shown in the main part of the manuscript. The full details can still be presented in the Appendix. Moreover, there is no explanation regarding the interpretation of the four tables in the Appendix. Please provide more explanation and/or a reading example for one of the tables.

L326: repeat again where the median and IQR values are shown "Using the median metrics along with their 25$^{th}$ and 75$^{th}$ percentiles (Table 3), the force curves idealized as Gaussians are shown in Figure 5.", or similar.

Section 3.3

In this section, shown in the Results, you almost exclusively describe your approach to derive force curves. Results are mentioned on L332-333 and in Figure 5. Should this be better split up in a part, which belongs to the Methods section, and a part, which is results? Consider moving L349-350 to this section, as it introduces results for a first time.

Figure 5: Consider showing the 6σ intervals in the figure, including the impact duration. You could also show the respective values for the peak force, making it much easier to grasp the key findings you derived from these curves.

Figure 5: You explain that you used the median and IQR values shown in Table 3 to derive the curves. I don't really understand why the idealized shoulder tap peaks at about 370 N, which seems very similar to the 373 N shown for the median in Table 3, rather than 343 N shown for the median. For wrist and elbow taps it is hard to judge from the figure whether this would also be the case. - Please explain why there is this difference.

4. Discussion: Obviously it is a matter of preference but consider re-ordering the Discussion section in four sections: (a) Explaining the variability in tapping force (currently in 4.3 and 4.5.2), Comparison with previous studies (currently in 4.1), Idealization of taps as Gaussian functions (currently in 4.3), and Implications for practitioners (currently in 4.2, 4.5).

Table 4: nice summary of the findings in the three studies

L340: Important to address these differences between studies. Consider mentioning that the differences between your study and Griesser et al. (2023) are even more astounding as about 62 participants participated in both studies at the EAWS General Assembly (Appendix-1).

L349-351: To me this is a finding. Consider moving to the respective Section 3.3, possibly after the sentence on L333.

L349: "We estimate the" – I really had to search for where you defined how you estimated this (a half-sentence on L316). Please make it easier for the reader (throughout the manuscript) to follow how you reached your results or conclusions, either by pointing the reader to respective parts of the manuscript, or by briefly repeating important facts, or by moving results closer to their definition.

Sect. 4.2 and 4.3: Consider changing the order of these sections and/or merging to one section where you first discuss potential reasons for variations, and then the implications of such variations. This would include discussing of body characteristics and gender, and possibly, the qualitative

observations (L461-466), which may all have partly contributed to the observed variations. It may also be possible to combine the implications of this variability with Sect. 4.5.2, where you again take up this topic.

L354-367: Consider rephrasing this section to something like "Variability in tapping force – implications for stability interpretations".

L355-359: From my perspective, this belongs to Section 3.1 as you present the results from a statistical test including p-values, which were not presented in the respective Results section.

L360-367: From my perspective, it is absolutely warranted to discuss the impact of the variability in tapping force with regard to interpreting stability test results (ECT or CT, in this case). But please provide a more in-depth discussion, combining your findings, how they translate to practice, and how your interpretation differs or aligns with previous work. Currently, it is hard for the reader to follow your statement: "especially in cases with potentially fatal outcomes". Explain how conducting a stability test is linked to potentially fatal outcomes. In that context I suggest to emphasize what (hopefully) is obvious "… no tests provide a definitive "go/no go" result. With accuracies of around 80%, tests are obviously not reliable enough to bet your life on them." (Birkeland et al., 2023) It is also not explained what your statement "our interpretation aligns with the principle of 'err on the side of caution'" specifically means (ignore the tapping number in test interpretation, I guess?). When discussing the test interpretation, it may also be worthwhile to state that in 21% of the cases two side-by-side ECT in the same snowpit didn't show the same propagation result regardless of tapping number (Techel et al. 2018). Similarly, Marienthal et al. (2023), who compared many previous studies and analyzed large data sets reported false-stable rates of 0-40% (previous studies) and 16-31% (new data), despite considering primarily the interpretation scheme based on fracture propagation. In other words, not only tapping force contributes to variability but there is a lot of (spatial) variability inherent in localized tests, with a high number of false-stable results, which again supports Birkeland's statement.

L379-380: This may be one explanation, but it is also possible that such a difference does not exist. – Please rephrase.

Sect. 4.5: important section to translate the findings to practice. Consider naming the section to "Implication for avalanche practitioners".

L419: You introduce the stuffblock test. Please add reference to the original paper and explain to the reader how this test works compared to CT or ECT.

L425: Consider changing from "limit" to "Revisiting the stability interpretation of CT and ECT"

L435: this simple, binary interpretation is widely accepted and not debated even by proponents of finer-scaled interpretation schemes (e.g., Techel et al., 2020 (p. 1949): "Quite clearly, whether a crack propagates across the entire column or not is the key discriminator between unstable and stable slope."). Consider/mention also that in your example, ECTP20 would be considered as "unstable" and ECTP24 as "intermediate" stability (but not as "stable") by Winkler et al. 2009. In other words, it is just a more gradual interpretation, with shades of grey in between rather than black and white what you seem to propose. Clearly, simplifying has advantages, but also comes at the cost of losing information. Therefore, providing a more in-depth discussion on the benefits/drawbacks of either approach is important when questioning whether three different loading steps are needed, and when proposing to revisit the ECT interpretation.

L461-466: I am not sure whether these qualitative observations are a limitation rather than an actual finding, which possibly may have impacted the force measurements? Consider moving this to the section where you discuss the variability between participants.

L495: Rephrase the "we" in the two questions to "avalanche practitioners" as done on L17-18 as the reader is not necessarily part of the "we".

**References**

Birkeland et al. (2023): https://arc.lib.montana.edu/snow-science/objects/ISSW2023_O9.04.pdf

Griesser et al. (2023): https://www.cambridge.org/core/journals/annals-of-glaciology/article/stress-measurements-in-the-weak-layer-during-snow-stability-tests/26DBD00E7309A4EF1C50F9FED88186F7

Techel et al. (2020): https://nhess.copernicus.org/articles/20/1941/2020/

Marienthal et al. (2023): https://arc.lib.montana.edu/snow-science/item/3009

Winkler and Schweizer (2009): https://www.wsl.ch/fileadmin/user_upload/WSL/Mitarbeitende/schweizj/Winkler_Schweizer_Stability_tests_CRST_2009.pdf

---

## Author Response (AR2)

Dear authors,

Please find attached some comments regarding your revised manuscript.

Thank you for your thorough review. We found your comments very constructive. We have tried our best to revise our manuscript in accordance with your suggestions. In the places where we disagree, we have tried to provide a thorough explanation of why. We hope that you will find that the responses sufficient for publication. If you should find that our responses are insufficient, we welcome further feedback on how to improve it.

I have three main points of critique:

- The presentation of results should be improved, to make it easier for the reader to grasp the findings of the study. Sometimes results are exclusively shown in the appendix with little or no information about these findings in the main part of the manuscript (e.g., L294-295), sometimes findings are presented for a first time in the Discussion (e.g., L355-359).

We have restructured the appendices into a new Table now included in the main results section. We have also moved L355-359 to the results section. See responses to specific comments below.

- It should be ascertained that readers who are not fully familiar with all the previous work (e.g., L419) or how the statistical analysis was done or can be interpreted (e.g., Appendix-4 to 7) can follow the line of argumentation. Please accommodate more of an outside view when revising by providing more of an explanation.

We have expanded on the statistical analysis and added a paragraph on how to interpret the data. See specific line comments below.

- It is clearly important to discuss the implication of the variability in tapping force regarding the stability interpretation of tests relying on hand tapping. However, this section needs much more of a balanced discussion combining findings (L364-365) with previous research on this topic.

We have addressed this in the line comments, hopefully you agree that the new paragraphs have a more balanced discussion.

In the following, please find some more detailed comments including on the three points mentioned before.

I hope this feedback helps to improve the manuscript.

**Comments:**

L12-13: Consider removing "Peak forces and loading rates are the metrics chosen to quantitatively compare the data." This statement seems redundant as the two metrics are used in the following sentence, and as no additional explanation is given in this sentence.

We have removed the sentence in the revised manuscript.

L13: Consider mentioning that peak force approximately doubles from one loading step to the next. This seems to be a key finding.

We have added the following: *and the peak force approximately doubles from one loading step to the next.*

L14: "Significant" overlap – Does this refer to statistical significance in the overlapping proportions? From L357 I understood that the distributions are significantly different, but there is no mention about the overlapping proportions being statistically different? Consider reserving "significant" only when referring

to statistical significance, and reword using something like "considerable", or similar. It may also pay to mention the approximate proportions of overlap in brackets.

We have replaced significant with considerable.

L16: introduce ECT abbreviation, when first used.

ECT is already introduced in L1: *This study examines the impact force applied from hand taps during Extended Column Tests (ECT), a common method of assessing snow stability.*

L22: Correct, but a bit strange that the concept of snowpack stability is introduced in the sense that it has been modelled in this way. What about: "Snowpack instability describes the propensity for a slope to avalanche (CITE). Failure initiation and crack propagation are key components of the avalanche release process (CITE)."
We have replaced the sentence as suggested.
L24-26: It is certainly correct that you state this but it interrupts reading the introduction. Consider rephrasing or moving to a different section.
We have moved this sentence to a footnote.

L30: Is this really the reason why stability tests were invented. Please cite the respective work where this is written. Otherwise, please rephrase more along the line that snow stability tests can support decision making in case of conditional stability (e.g., Birkeland et al, 2023).
We have rephrased the sentence as suggested.

L34: Consider to start the sentence with "In contrast, in situations…"
We have rephrased the sentence as suggested.

L35: Consider explaining that this division is based on informational entropy, by emphasizing that the mentioned signs (L34) allow direct interpretation of instability (class 1), while stability tests provide more indirect information (class 2).
We have rephrased this paragraph trying to reflect the comments provided.

L69: "simulate portions of" – maybe rephrase to "reflect the"
We have rephrased the sentence as suggested.

L72: "component of the ECT" – consider adding the CT: "component of CT and ECT"
We have replaced ECT with CT and ECT throughout the paragraph.

L127: Please rephrase to accommodate that Griesser et al. (2023) not only performed tests in an indoord setting but also in the field.
We have rephrased the sentence to highlight the field component of their study: *Furthermore, Griesser et al. (2023) performed stress measurements during CTs in the field and investigated the effects of body characteristics such as weight and height.*

L134: "improved measurement device" – please specify relative to what this improvement is intended to be.
We have specified that the improvement is the increased sampling rate which allow us to sample the entire impact curve of hand taps.

L199: Introduce EAWS when first using this abbreviation.
We have rephrased this sentence as suggested.

L212-213: Consider removing, as it is stated in the sentence before.

We added this sentence as a response to reviewer comment from Simenhois which wanted us to explicit state this. We have now removed the sentence again as we agree with reviewer comment from Techel.

L258-259: Consider removing "After this second… ".
We have removed the sentence as suggested.

Sect. 2.4: Consider reordering this paragraph in the way you present the results: (1) methods used to compare tapping force, (2) methods used to analyze explanatory factors. Please explain in a little more detail what ANOVA actually compares and provide a reference. (for instance, I had to look it up as I rarely use this technique). Consider bringing the second sentence first, then the first sentence. Provide references to all methods and tests.
We have rephrased, expanded on what an ANOVA does, and added a citation.
*We tested height, weight, gender, and geographic region to understand the underlying factors influencing hand-tap loading using ordinary least squares (OLS) regression models. The peak force was the dependent variable in these models. To compare hand-tap loading at different loading steps, we conducted a one-way ANOVA. This analysis assessed whether the mean impact forces were statistically different during wrist, elbow, and shoulder taps. ANOVA, or Analysis of Variance, compares the means of three or more groups to determine if at least one group's mean is significantly different from the others (Fisher, 1970). All analyses were considered statistically significant at p-values below 0.05.*

*Fisher, R. A. (1970). Statistical methods for research workers. In Breakthroughs in statistics: Methodology and distribution (pp. 66-70). New York, NY: Springer New York.*

L271: Is "trend" the right word for your analysis? Consider using an alternative word.
We have changed the subsection title from "Trends and variability by individual tappers" to "Peak force and loading rate."

L272-277: this is essentially a repetition of Table 2. Consider shortening, highlighting one or key facts, and referring the reader to the table for more information.
We have shortened this section by removing most of the repetitive information.

L276: Consider removing "~0.0%" as you state that it happened once (in about 8000 tests).
Removed in response to L272-277.

L281: remove "(outliers removed using 1.5 times IQR)" as this is already introduced on L272-274
Removed as suggested.

L283: "(wrist, elbow, shoulder)" – consider deleting, as the loading steps are explained several times before
Removed as suggested.

L285: "each load step (i.e. loading step)" – consider removing one of these
Removed as suggested.

Table 3: Explain abbreviation "Std." when first being used in the manuscript.
We have removed the abbreviation.

L289-292: This is essentially the same as the caption of Figure 4. Consider removing from text or from caption.
We have shortened the text in L289-292.

L294-295: Why make this statement or add this table shown in Appendix-3 if you don't explain what is shown? Moreover, I would move this way of looking at the data in a more prominent way to the Results section, as it provides another way at interpreting the data, with results which allow a simple way of

summarizing the proportions of agreements and disagreements. - I suggest incorporating this into the main part of the manuscript, but at least to rephrase the current statement giving the reader at least the essence of what is shown in Appendix-3. For instance, findings like "defining nonoverlapping band-widths of tapping force based on the distribution of tapping forces shown in Table Table 3 and Figure 4 showed that between 49% and 54% of the taps of a loading step were within this range. However, a share of about 1% of the taps applied a force corresponding to the range of two loading steps higher or lower". or similar, could be mentioned.

We have added Appendix-3 to the results section as suggested.

L305: "cannot explain the bulk of the variance" – Please be more specific with regard to how much these factors could explain or consider rephrasing.

We have rewritten entire section 3.2 expanding on this.

L303: Consider renaming the section title to something like "Explanatory factors impacting peak force" or similar.

We have renamed the section as suggested.

L303-309: For the reader it is hard to follow the line of argumentation as the reader is being referred four times to different tables in the Appendix. I suggest summarizing these four tables in a compact table shown in the main part of the manuscript. The full details can still be presented in the Appendix. Moreover, there is no explanation regarding the interpretation of the four tables in the Appendix. Please provide more explanation and/or a reading example for one of the tables.

We have removed the OLS appendices and included a new Table in the revised manuscript.

L326: repeat again where the median and IQR values are shown "Using the median metrics along with their 25th and 75th percentiles (Table 3), the force curves idealized as Gaussians are shown in Figure 5.", or similar.

We have rephrased the sentence as suggested.

Section 3.3

In this section, shown in the Results, you almost exclusively describe your approach to derive force curves. Results are mentioned on L332-333 and in Figure 5. Should this be better split up in a part, which belongs to the Methods section, and a part, which is results? Consider moving L349-350 to this section, as it introduces results for a first time.

We agree that much of section 3.3 is better suited for in a methods section. We have moved this portion of section 3.3 to a new section 2.5.

Figure 5: Consider showing the 6σ intervals in the figure, including the impact duration. You could also show the respective values for the peak force, making it much easier to grasp the key findings you derived from these curves.

We feel including peak force values and 6σ values on the figure would be too cluttered. We have moved the result of the 6σ calculation of duration to section 3.3 and put the specific values of 21 ms, 14 ms, and 11 ms for the wrist elbow and shoulder, rather than the previous rounded estimations. So, they are now conveniently placed for the reader to visualize with Figure 5. The median peak force values are already shown in Table 3 and are not included here to avoid redundancy.

Figure 5: You explain that you used the median and IQR values shown in Table 3 to derive the curves. I don't really understand why the idealized shoulder tap peaks at about 370 N, which seems very similar to the 373 N shown for the median in Table 3, rather than 343 N shown for the median. For wrist and elbow taps it is hard to judge from the figure whether this would also be the case. - Please explain why there is this difference.

Good catch, this was a mistake on our end. In the previous version of the manuscript, we had used mean peak force and loading rate values to generate the center curves, rather than median values. The values

used for shading (IQR) were correct, but the center line was incorrect. We have revised the figure to now use the median values.

The linear impulses were also originally calculated with the mean, rather than median, values. This led to slight changes in calculated linear impulse values which are now correct in the revised manuscript.

The script to generate this figure and calculate durations and linear impulses can be found in the Open Science Framework repository.

4. Discussion: Obviously it is a matter of preference but consider re-ordering the Discussion section in four sections: (a) Explaining the variability in tapping force (currently in 4.3 and 4.5.2), Comparison with previous studies (currently in 4.1), Idealization of taps as Gaussian functions (currently in 4.3), and Implications for practitioners (currently in 4.2, 4.5).
Our preference is to keep the discussion as is except for changing the order of section 4.2 and 4.3 as suggested. We believe that the comparison with previous studies in relation to our results should come first.

Table 4: nice summary of the findings in the three studies
Thank you.

L340: Important to address these differences between studies. Consider mentioning that the differences between your study and Griesser et al. (2023) are even more astounding as about 62 participants participated in both studies at the EAWS General Assembly (Appendix-1).
If we compare our data from EAWS with the rest, there is no statistical valid difference (i.e. Griesser et al. data is representable of ours). It is simply a result of different sampling rate. We have added these sentences to the end of the paragraph to highlight that differences are likely due to the different sampling rate: *We have measured the 62 participants from Griesser et al. (2023) in parallel with our own measurement device, and the measurements are similar to the rest of our samples. This comparison suggests that the differences are likely due to the difference in sampling rate.*

L349-351: To me this is a finding. Consider moving to the respective Section 3.3, possibly after the sentence on L333.
We have moved this sentence to the end of L333 as suggested.

L349: "We estimate the" – I really had to search for where you defined how you estimated this (a half-sentence on L316). Please make it easier for the reader (throughout the manuscript) to follow how you reached your results or conclusions, either by pointing the reader to respective parts of the manuscript, or by briefly repeating important facts, or by moving results closer to their definition.
We have moved the loading duration results to section 3.3. We also added $6\sigma$ in parentheses after the word "duration" to remind the reader how it is calculated and referred them to section 2.5. The sentence now reads: *We estimate the median loading duration ($6\sigma$, section 2.5) of the impact curve to be 21 ms for the wrist, 14 ms for the elbow, and 11 ms for the shoulder.*

Sect. 4.2 and 4.3: Consider changing the order of these sections and/or merging to one section where you first discuss potential reasons for variations, and then the implications of such variations. This would include discussing of body characteristics and gender, and possibly, the qualitative observations (L461-466), which may all have partly contributed to the observed variations. It may also be possible to combine the implications of this variability with Sect. 4.5.2, where you again take up this topic.
We have changed the order of Section 4.2 and 4.3 and added L461-466 as suggested. We have selected to keep section 4.5.2 as it summarizes the implications of variability and how that applies to our two ideas for further discussion.

L354-367: Consider rephrasing this section to something like "Variability in tapping force – implications for stability interpretations".

We have rephrased this subsection as suggested.

L355-359: From my perspective, this belongs to Section 3.1 as you present the results from a statistical test including p-values, which were not presented in the respective Results section.
We have moved these sentences to Section 3.1 as suggested.

L360-367: From my perspective, it is absolutely warranted to discuss the impact of the variability in tapping force with regard to interpreting stability test results (ECT or CT, in this case). But please provide a more in-depth discussion, combining your findings, how they translate to practice, and how your interpretation differs or aligns with previous work. Currently, it is hard for the reader to follow your statement: "especially in cases with potentially fatal outcomes". Explain how conducting a stability test is linked to potentially fatal outcomes. In that context I suggest to emphasize what (hopefully) is obvious "… no tests provide a definitive "go/no go" result. With accuracies of around 80%, tests are obviously not reliable enough to bet your life on them." (Birkeland et al., 2023) It is also not explained what your statement "our interpretation aligns with the principle of 'err on the side of caution'" specifically means (ignore the tapping number in test interpretation, I guess?). When discussing the test interpretation, it may also be worthwhile to state that in 21% of the cases two side-by-side ECT in the same snowpit didn't show the same propagation result regardless of tapping number (Techel et al. 2018). Similarly, Marienthal et al. (2023), who compared many previous studies and analyzed large data sets reported false-stable rates of 0-40% (previous studies) and 16-31% (new data), despite considering primarily the interpretation scheme based on fracture propagation. In other words, not only tapping force contributes to variability but there is a lot of (spatial) variability inherent in localized tests, with a high number of false-stable results, which again supports Birkeland's statement.
We have rephrased this entire paragraph; we hope that you agree that we have addressed your concerns.

*It is widely agreed that whether a crack propagates across the entire column or not is the key discriminator between unstable and stable slopes (Techel et al., 2020). However, both Winkler and Schweizer (2009) and Techel et al. (2020) show that the number of taps provides additional information, allowing a more refined distinction between results related to stable and unstable conditions. Techel et al. (2020) found the optimal threshold between ECTP20 and ECTP22, which aligns with the ECTP21 threshold suggested by Winkler and Schweizer (2009). Moving away from a binary classification came at the cost of introducing intermediate stability classes (Techel et al., 2020).*

*These new intermediate stability class definitions rely heavily on the tap number when failure occurs. Variability in the applied force-time curves likely leads to variability in test results, particularly regarding the number of taps required to induce weak layer failure. It is important to emphasize that no tests offer a definitive "go/no go" result. With accuracies of around 80%, these tests are not reliable enough to be the main factor in our slope scale decision-making (Birkeland et al., 2023).*

*We found the three loading steps to be statistically different; this aligns with the results from Griesser et al. (2023), which highlight this as a positive outcome and that the CT and ECT hand-tap procedure is somewhat reliable. Despite the statistical differences in each loading step, we question the application of average results to individual cases. The main difference in our argument lies in relying solely on mean statistics to develop impact force thresholds used by individuals. There is significant overlap between the 25th-75th percentile ranges of force applied during elbow taps with those of wrist and shoulder taps, where ~18% and ~26% of the data for elbow taps overlap with wrist and shoulder taps, respectively (Table 4). These overlaps have practical significance in real-world applications.*

L379-380: This may be one explanation, but it is also possible that such a difference does not exist. – Please rephrase.
We have rephrased the sentence: *The lack of significant findings might be attributed to our limited predictive capability from the small sample size in a statistical context (n=286), or that there are no differences to be found.*

Sect. 4.5: important section to translate the findings to practice. Consider naming the section to "Implication for avalanche practitioners".

We have renamed the Section as suggested.

L419: You introduce the stuffblock test. Please add reference to the original paper and explain to the reader how this test works compared to CT or ECT.

We have added the citation and rephrased to elaborate on how the test works.

Another solution could be to develop a tool that ensures consistent impact force, like the stuffblock test (Johnson and Birkeland, 1998). The test involves filling a nylon sack with 4.5 kg of snow and dropping it in increments of 10 cm. However, this test type of loading has its challenges.

L425: Consider changing from "limit" to "Revisiting the stability interpretation of CT and ECT"

We have rephrased Section 4.5.2 as suggested.

L435: this simple, binary interpretation is widely accepted and not debated even by proponents of finer-scaled interpretation schemes (e.g., Techel et al., 2020 (p. 1949): "Quite clearly, whether a crack propagates across the entire column or not is the key discriminator between unstable and stable slope."). Consider/mention also that in your example, ECTP20 would be considered as "unstable" and ECTP24 as "intermediate" stability (but not as "stable") by Winkler et al. 2009. In other words, it is just a more gradual interpretation, with shades of grey in between rather than black and white what you seem to propose. Clearly, simplifying has advantages, but also comes at the cost of losing information. Therefore, providing a more in-depth discussion on the benefits/drawbacks of either approach is important when questioning whether three different loading steps are needed, and when proposing to revisit the ECT interpretation.

We have added that ECTP20 would be considered as "unstable" and ECTP24 as "intermediate" stability (but not as "stable") by Winkler et al. 2009 as suggested. We have elaborated on the whether the three different loading steps are needed in response to L360-367.

L461-466: I am not sure whether these qualitative observations are a limitation rather than an actual finding, which possibly may have impacted the force measurements? Consider moving this to the section where you discuss the variability between participants.

We have moved this paragraph to Section 4.2 as suggested.

L495: Rephrase the "we" in the two questions to "avalanche practitioners" as done on L17-18 as the reader is not necessarily part of the "we".

We have rephrased the sentences as suggested.

**References**
Birkeland et al. (2023): https://arc.lib.montana.edu/snow-science/objects/ISSW2023_O9.04.pdf
Griesser et al. (2023): https://www.cambridge.org/core/journals/annals-of-glaciology/article/stressmeasurements-in-the-weak-layer-during-snow-stabilitytests/26DBD00E7309A4EF1C50F9FED88186F7
Techel et al. (2020): https://nhess.copernicus.org/articles/20/1941/2020/
Marienthal et al. (2023): https://arc.lib.montana.edu/snow-science/item/3009
Winkler and Schweizer (2009): https://www.wsl.ch/fileadmin/user_upload/WSL/Mitarbeitende/schweizj/Winkler_Schweizer_Stability_tests_CRST_2009.pdf